# Selective peptide bond formation via side chain reactivity and self-assembly of abiotic phosphates

Arti Sharma[1,2], Kun Dai[3], Mahesh D. Pol[2,3], Ralf Thomann[1,4], Yi Thomann[1], Subhra Kanti Roy[2] & Charalampos G. Pappas [1,2,3] ✉

In the realm of biology, peptide bonds are formed via reactive phosphate-containing intermediates, facilitated by compartmentalized environments that ensure precise coupling and folding. Herein, we use aminoacyl phosphate esters, synthetic counterparts of biological aminoacyl adenylates, that drive selective peptide bond formation through side chain-controlled reactivity and self-assembly. This strategy results in the preferential incorporation of positively charged amino acids from mixtures containing natural and non-natural amino acids during the spontaneous formation of amide bonds in water. Conversely, aminoacyl phosphate esters that lack assembly and exhibit fast reactivity result in random peptide coupling. By introducing structural modifications to the phosphate esters (ethyl *vs.* phenyl) while retaining aggregation, we are able to tune the selectivity by incorporating aromatic amino acid residues. This approach enables the synthesis of sequences tailored to the specific phosphate esters, overcoming limitations posed by certain amino acid combinations. Furthermore, we demonstrate that a balance between electrostatic and aromatic stacking interactions facilitates covalent self-sorting or co-assembly during oligomerization reactions using unprotected N-terminus aminoacyl phosphate esters. These findings suggest that self-assembly of abiotic aminoacyl phosphate esters can activate a selection mechanism enabling the departure from randomness during the autonomous formation of amide bonds in water.

Phosphates and phosphate esters are fundamental to key biological functions, maintaining structural integrity, driving energy transfer, and enabling information storage within living organisms[1]. Their amphiphilic nature allows them to form assemblies, which play a key role in cellular structures, such as microtubules[2] and membranes[3]. Phospholipids, are a prime example of such amphiphilic molecules, forming the structural backbone of biological membranes. The combination of hydrophobic tails and polar phosphate head groups enables these molecules to self-assemble into larger structures, influencing a wide array of membrane properties and functions[4]. Such processes allow spatial coupling of metabolic pathways[5,6], protect against parasites in scenarios of competing entities[7], and offer control over internal microenvironment[8]. Beyond their amphiphilic role, phosphates are also central to macromolecular synthesis[9]. In protein synthesis, amino acids are activated by phosphate-rich molecules like adenosine triphosphate, forming aminoacyl adenylates that are organized within

[1]FIT – Freiburg Center for Interactive Materials and Bioinspired Technologies, University of Freiburg, Freiburg, Germany. [2]Institute of Organic Chemistry, University of Freiburg, Freiburg, Germany. [3]DFG Cluster of Excellence livMatS @FIT – Freiburg Center for Interactive Materials and Bioinspired Technologies, University of Freiburg, Freiburg, Germany. [4]Freiburg Materials Research Center (FMF), University of Freiburg, Freiburg, Germany.
✉e-mail: charalampos.pappas@livmats.uni-freiburg.de

the endoplasmic reticulum membrane. These processes highlight the dual role of phosphates, serving both structural functions and regulating reactivity, which are essential for ensuring the accuracy of biomolecular assembly and synthesis[10].

Outside of biology, regulating self-assembly and reactivity has gained significant attention as a strategy to control the behavior of chemical systems[11–13]. Assemblies made of various compound classes, such as fatty acids[14], peptides[15], nucleobases[16], and chimeric structures[17,18] have shown to affect the dynamics, the assembling pathway, and the supramolecular architecture of different types of structures[19]. Particular to peptides, short sequences featuring aromatic and aliphatic amino acid residues[20,21] have been used to trigger the formation of spherical aggregates, highlighting the versatility of amino acid side chains and the different types of non-covalent interactions that are involved in constructing dynamic phase changes[22,23]. Moreover, systems composed of oppositely charged amino acid residues have shown to induce phase seperation[24,25], which in turn gave rise to responsive behavior[26] and in the selective uptake of guest molecules[27]. In these systems, structure formation is mainly triggered by utilizing phosphate-arginine[28] or phosphate-lysine[29] electrostatic interactions. Chemical reactions, including imine formation[30], aldol condensation[31], and anhydride formation[32], have also been successfully coupled with assembly formation, further illustrating how compartmentalized structures can impact the properties of synthetic systems. However, chemical systems that investigate the way in which aggregation of activated precursors into compartments[33] can affect structure, reactivity and induce selection without the need of enzymatic machinery remain rare. In the context of amide bond formation[34–36], selective peptide coupling has been demonstrated using redox-active coacervates[37], which facilitated controlled reactivity by creating microenvironments that stabilize intermediates and enhance specificity. Similarly, surfactant aggregates[38] have been shown to drive spontaneous polypeptide formation from aminoacyl adenylates[39,40], highlighting the role of self-assembled systems in oligomerization reactions. In other strategies, wet-dry cycles[41] and metal ions[42] have been employed to induce peptide elongation in aqueous media, however these conditions are often incompatible with the formation of stable self-assembled structures. Additionally, enzymatic methods[43], thioester formation[44], and the use of N-carboxy anhydrides[45] have allowed for the coupling of specific amino acid residues, yet these strategies face challenges in solubility, reactivity control and selectivity of self-assembling motifs.

In this work, we use aminoacyl phosphate esters, synthetic analogs of biological aminoacyl adenylates, which possess distinct structural components (N-terminus, amino acid side chain, and phosphate ester) that can be tailored to fabricate activated amino acid precursors capable of self-assembly. The reactivity of these esters is modulated by the nature of the amino acid side chains, with a clear distinction between aliphatic and aromatic residues. Self-assembly of phosphate esters promotes selective incorporation of positively charged residues from mixtures of amino acids and peptides, enabling a departure from randomness during the spontaneous formation of amide bonds. This mechanism also extends to mixtures containing non-natural amino acids. In contrast, aminoacyl phosphate esters lacking structural organization display rapid reactivity, resulting in random peptide coupling. By introducing further modifications to the phosphate esters, while maintaining their ability to form organized structures, we are able to tune selectivity by incorporating aromatic residues in the resulting sequences. Finally, we demonstrate that the self-assembly of aminoacyl phosphate esters drives covalent self-sorting during oligomerization reactions involving N-terminus-free activated amino acids.

## Results
### Design of aminoacyl phosphate esters capable of aggregation
We have previously demonstrated that aminoacyl phosphate esters led to the spontaneous and selective formation of peptide oligomers in an aqueous environment as a result of autonomous phase changes[46]. The amino acid residues in the structure of aminoacyl phosphate esters dictated the length of oligomerization and the composition in different phases. Thus, selectivity in these systems was achieved via the self-assembly of the resulting homo-oligomers. Herein, we focus on achieving selectivity through reactivity control and self-assembly of aminoacyl phosphate precursors. Our approach focuses on designing systems that examine how activated amino acids can: (1) integrate structural elements in their structure that promote self-assembly, (2) regulate structure and reactivity, enabling the selective formation of peptide bonds in water, and (3) induce covalent self-sorting or co-assembly through oligomerization reactions. In order to satisfy these criteria, we used aminoacyl phosphate esters, where the phosphate esters enhance the solubility of hydrophobic amino acids residues in an aqueous environment[47]. Different parts of the molecule, including the N-terminus, amino acid side chains, and phosphate esters were engineered to enhance or disrupt electrostatic and aromatic stacking interactions governing assembly. Specifically, at the N-terminus, we used the tert-butyloxycarbonyl (Boc) group, a commonly used Nα-amino protecting group in peptide synthesis, which has also been utilized to trigger the formation of assemblies from short peptide sequences[48]. The amino acid side chains were modified to include both aliphatic and aromatic moieties. For the aromatic residues, functionalization was introduced at the *para* position of phenylalanine, incorporating bulky groups such as fluorenylmethoxycarbonyl (Fmoc) and benzoyl. These modifications are commonly used in peptide-based chemical systems to promote assembly formation[49–51]. Additionally, we incorporated charged moieties at the *para* position of phenylalanine residues, such as guanidine and carboxylic acid[52], to investigate how the combination of electrostatic and aromatic stacking interactions can influence oligomerization. The phosphate esters were equipped with an ethyl or a phenyl moiety. Molecules with a free N-terminus are classified as series 1, while those with a Boc-protected group are categorized as series 2 (Fig. 1).

We prepared mixtures of amino acids and dipeptides in order to investigate selective peptide coupling. Amino acid mixture I was consisted of an aromatic (phenylalanine-F), an aliphatic (leucine-L), a hydrophilic (serine-S), a positively (arginine-R), and a negatively charged (acid-D) amino acid residue. We use the single-letter code for the amino acids throughout this work. Additional mixtures involved non-natural amino acids featuring charged moieties on the *para* position of phenylalanine (amino acid mixture II). Finally, mixtures incorporating dipeptide sequences, including DD, DR, DF, and DW (where W represents tryptophan) were prepared in order to explore how a short sequence can affect the coupling with aminoacyl phosphate esters.

### Departure from randomness
We started by investigating the behavior of the aminoacyl phosphate ester **Boc-FEP (2a)** in an aqueous environment (borate buffer 0.6 M pH 9.1). Upon hydrolysis, the corresponding carboxylic acid (**Boc-F-OH/2a-OH**) and ethyl phosphate (EP) is formed. We determined the half-life of **2a** to be 2.2 h, with the system maintaining macroscopic transparency throughout the reaction, indicating that both the aminoacyl phosphate ester and the resulting carboxylic acid remained in solution (Supplementary Fig. 1A). Next, we explored peptide bond formation using **2a** with various amino acids in individual reactions, such as D, S, L, R, F, W, as well as K (lysine), H (histidine), P (proline), and I (isoleucine). Peptide bond formation reached 70–80% for most tested amino acids, as confirmed by ultraperformance liquid chromatography-mass spectrometry (UPLC-MS), indicating a highly efficient and spontaneous process[53,54]. These results show that slight differences in the pKa of the α-amino groups of the amino acids had minimal effect on the efficiency of the coupling reaction. Peptide conversions were determined after the completion of the reaction,

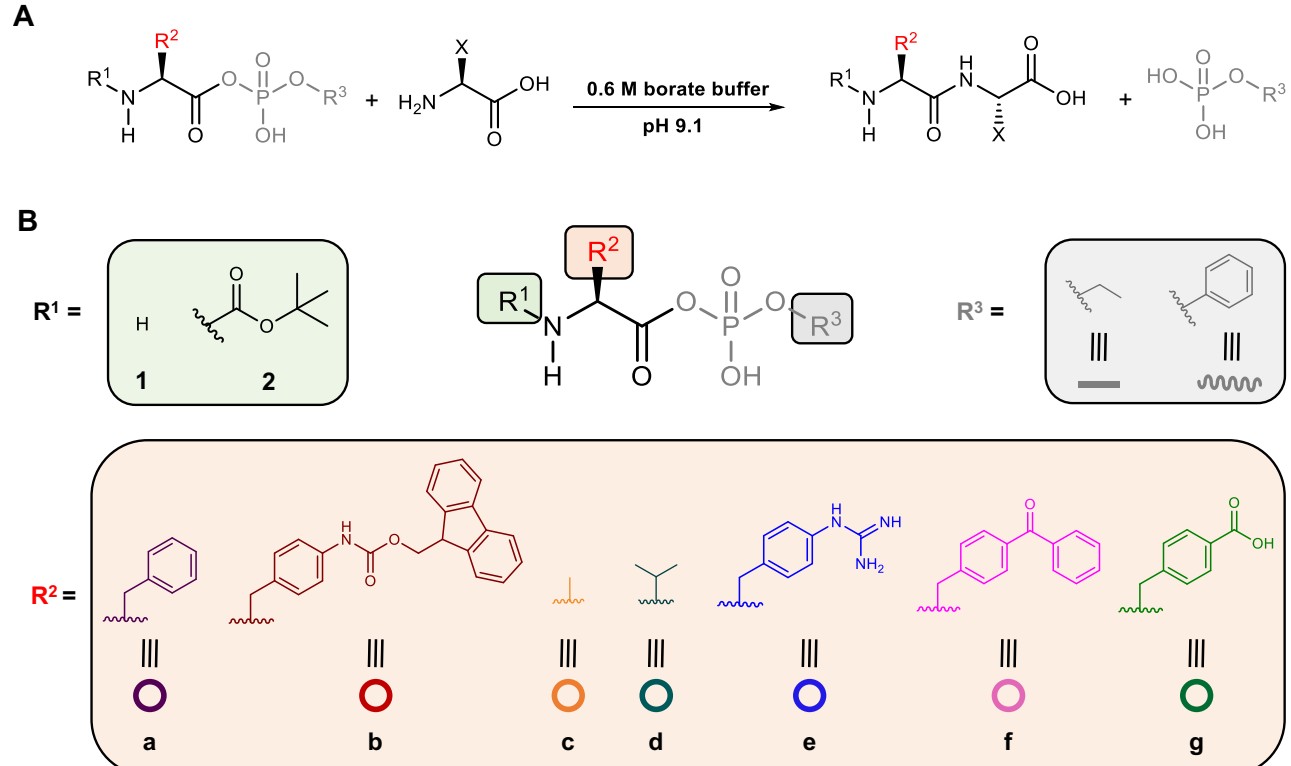

**Fig. 1 | Aminoacyl phosphate ester-mediated peptide coupling in aqueous media. A** Peptide bond formation between aminoacyl phosphate esters and amino acids in borate buffer (0.6 M, pH 9.1). **B** Chemical structures of aminoacyl phosphate esters, highlighting key components: the N-terminus (free -H or protected as tert-butoxycarbonyl (Boc), represented as R¹), amino acid side chains (R² = a–g),

and phosphate ester groups (R³ = ethyl or phenyl). These structural variations enable a systematic investigation of their influence on peptide bond formation and reaction kinetics. Direct hydrolysis of the aminoacyl phosphate esters which competes with peptide bond formation is not depicted in the reaction.

which occurred over 60 min. Coupling was also achieved with secondary amines, such as proline, with up to 85% conversion when using **2a** as the acyl donor. One exception to this general trend was observed with aspartic acid, where hydrolysis of the aminoacyl phosphate ester dominated over peptide bond formation. This may be attributed to electrostatic repulsion between the negatively charged carboxylate groups of aspartic acid and the EP moiety of **2a**. We then conducted competition experiments using mixtures of five amino acids (D, S, L, R, F) at a 1:1 molar ratio with **2a**. These experiments revealed random incorporation of the amino acids into peptides, accompanied with distinct formation of the hydrolysis product (Supplementary Fig. 2A). In order to mitigate hydrolysis and promote peptide bond formation, we increased the ratio of aminoacyl phosphate ester to amino acid mixture I to 1:5. In these experiments, we observed a slight increase in peptide coupling with R, reaching 35%. In the same mixture, F showed a coupling efficiency of 23%, while S reached almost 20%. There were no significant changes observed when libraries were prepared at different pH (Supplementary Fig. 2B, C), suggesting that product distribution was mainly influenced by the intrinsic reactivity of the phosphate ester. The UPLC-MS chromatograms and peptide bond conversions for the systems containing **2a** and amino acid mixtures are available in the Supplementary Information (Supplementary Figs. 3–21).

Having observed random incorporation from **2a**, we aimed to enhance selectivity by slowing down the system's reactivity (Fig. 2A). Two different approaches were followed in parallel. The first examined the self-assembly propensity of the aminoacyl phosphate esters, while the second focused on modifications to the amino acid side chains. Replacing **2a** with **2b** (**Boc-F(4-NH-Fmoc)EP**), the half-life of the activated amino acid was increased and found to be 99 h (Supplementary Fig. 1B). Unlike **2a**, the dissolution of **2b** resulted in the formation of foamy structures (inset digital photos in Fig. 2B), which

corresponded to nanoscale assemblies. Structure formation was confirmed using cryo-transmission electron microscopy (Cryo-TEM), revealing spherical aggregates ranging from 6 to 19 nm in size. Over time these aggregates underwent a supramolecular reconfiguration, forming a hydrogel, which corresponded to the carboxylic acid derivative (**2b-OH**) (Fig. 2B). These findings were further supported by confocal microscopy and dynamic light scattering (DLS) (Supplementary Figs. 22 and 23). The reconfiguration process was accompanied by a reduction in fluorescence intensity, suggesting changes in the arrangement of the fluorophores (Supplementary Fig. 24)[49]. From random peptide coupling between **2a** and amino acid mixture I (Fig. 2C), we observed selective peptide bond formation with R (60%) when **2b** was used (Fig. 2D). Other amino acids from the mixture yielded less than 10% coupling with **2b**. Peptide conversions were monitored via UPLC and assessed after the completion of the reaction, which occurred over 48 h. To investigate the aggregation dynamics following selective peptide bond formation with R, the system was further characterized. Confocal microscopy of a mixture containing 10 mM **2b** and 10 mM R in 0.6 M borate buffer (pH 9.1) revealed a transition from smaller aggregates formed by **2b** alone to larger assemblies, indicating that the peptide coupling product between R and **2b** promoted aggregate growth (Supplementary Fig. 25). This observation was supported by fluorescence spectroscopy, which showed a decrease in intensity and a red shift, suggesting changes in the arrangement of the aromatics (Supplementary Fig. 26). Similar aggregates were visualized using confocal microscopy in amino acid mixture I with **2b**, further confirming that selective peptide bond formation with R promotes aggregation under competitive conditions (Supplementary Fig. 27). Ratio and pH-dependent experiments of **2b** with amino acid mixture I revealed a direct competition between coupling with R and hydrolysis of the activated amino acid

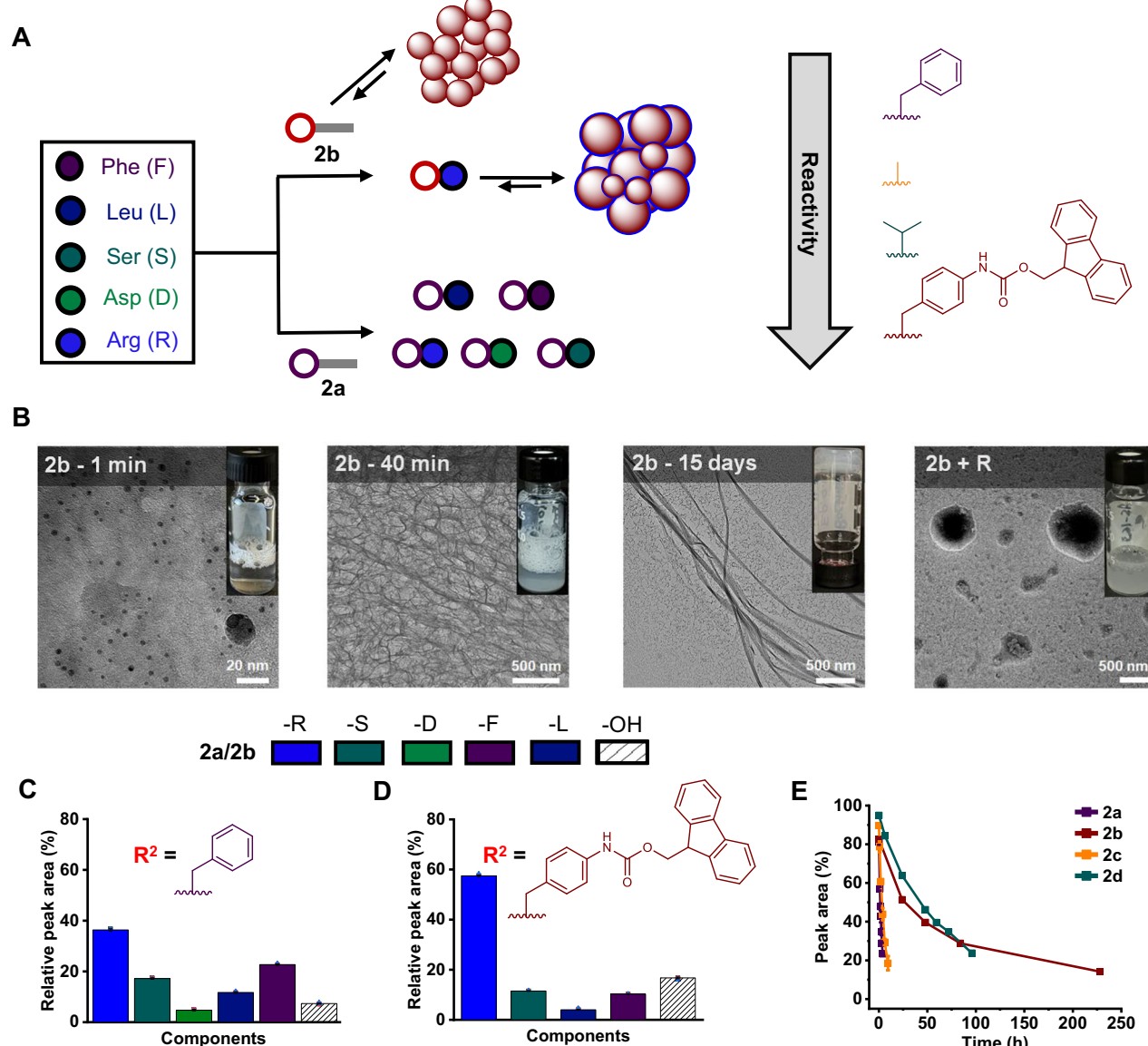

**Fig. 2 | Departure from randomness through self-assembly of aminoacyl phosphate esters. A** Peptide bond formation from **2a** or **2b** with amino acid mixtures highlighting the reactivity trend of **2a, 2b, 2c,** and **2d**. **B** Cryo-transmission electron microscopy images (from left to right) of 10 mM **2b** immediately after dissolving, after 40 min, after 15 days, and upon mixing **2b** with **R**. Peptide bond formation between (**C**) **2a** and (**D**) **2b** with amino acid mixture I. The concentration used for **2a** or **2b** was 10 mM, while the total concentration for the amino acids in the mixture was 50 mM (10 mM each). Striped bars indicate hydrolysis products (**2a-OH** or **2b-OH**). Peptide coupling yields were measured at the end of each reaction: for **2a**, the reaction completes in 30 min, whereas **2b** required 48 h. **E** Time-dependent hydrolysis of aminoacyl phosphate esters. Error bars represent the standard deviation of three independent experiments.

(Supplementary Fig. 28). Building on the selective coupling with R, we explored the behavior of lysine (K), another positively charged amino acid with distinct reactivity attributed to its flexible aliphatic side chain terminating in an ε-amine group. Replacing R with K in mixture I resulted in peptide bond formation with K, demonstrating the important role of electrostatic interactions in promoting selective coupling (Supplementary Figs. 29A, 30, and 31A). It has been observed that lysine undergoes a two-step coupling process, where initial bond formation occurs with one of its primary amines, followed by subsequent coupling involving a second reactive amine. As expected, the use of trimethyl lysine eliminated side reactions from the ε-amine, while still maintaining selective peptide bond formation (Supplementary Figs. 29B, 31B, and 32–34). In competition experiments between R and K using **2b**, we were unable to separate the coupling products using UPLC. Fluorescence microscopy revealed the absence of the typical spherical aggregates observed with high yields of arginine

coupling products, suggesting that lysine was also incorporated (Supplementary Fig. 35). To overcome the analytical challenges with R and K, we introduced ornithine (O) and diaminopropionic acid (Dpr). Competition experiments between R and O, as well as R and Dpr, were carried out with both **2a** and **2b**. The results indicated a mixture of products involving both amino acids (Supplementary Figs. 36–45). Additionally, O and Dpr were tested by replacing K in amino acid mixture I, and similar trends in peptide coupling were observed, further supporting the role of side chain interactions in determining the outcome of these reactions (Supplementary Figs. 46–55). Further examining the impact of different side chain configurations, we introduced proline in the original amino acid mixture I. However, this modification did not alter the coupling behavior in either the **2a** or **2b** containing libraries (Supplementary Figs. 56–60). The importance of structural organization of aminoacyl phosphate esters was evident in experiments conducted in a co-solvent environment. In presence of

80% acetonitrile (ACN), random peptide coupling was observed due to the disruption of aggregation. Furthermore, high concentrations of salts, such as sodium sulfate, induced random mixtures, suggesting a structure-breaking effect (Supplementary Figs. 61–64). Experiments with other salts primarily resulted in hydrolysis, preventing conclusions from the Hofmeister series[55] (Supplementary Fig. 65 and Supplementary Table 1). The preference for peptide bond formation with positively charged amino acids was further confirmed by using non-natural amino acids (amino acid mixture II), where phenylalanine residues were functionalized with ionic groups at the *para* position. While samples containing **2a** and amino acids from mixture II led to random coupling, mixtures with **2b** selectively formed peptide bonds with the F(4-guanidine) amino acid residue (Supplementary Figs. 66–71).

Having demonstrated the role of aromatic amino acids in the structure of phosphate esters in promoting aggregation and peptide coupling, we were prompted to explore the impact of other side chains on peptide bond formation with amino acid mixture I. Thus, we synthesized two additional derivatives, Boc-AEP (**2c**) and Boc-VEP (**2d**), to investigate how aliphatic residues could influence the process. Boc-AEP (**2c**) exhibited a moderate half-life of 4 h similar to Boc-FEP (**2a**), whereas Boc-VEP (**2d**) showed a significantly longer half-life of 49.5 h, highlighting the stabilizing effect of the bulkier aliphatic side chain (Fig. 2E, Supplementary Fig. 1c, d). These experiments revealed distinct differences in reactivity between aromatic and aliphatic amino acids. Aromatic side chains in aminoacyl phosphate esters exert inductive effects, increasing the electrophilicity of the carbonyl group, making it more prone to hydrolysis. However, in the presence of aggregates, this susceptibility is reduced as the activated amino acids become shielded from external nucleophiles, including water. **2c** showed a product distribution similar to **2a** with amino acid mixture I, as both displayed comparable reactivity (Supplementary Figs. 72–74). Interestingly, when alanine was replaced with valine (**2d**), reactivity significantly decreased, yet this led to enhanced peptide coupling with R (Supplementary Figs. 75–77). Time-dependent analysis of the libraries (Supplementary Fig. 78) revealed that **2b** exhibited slower kinetics compared to all other derivatives but demonstrated the highest selectivity. Although **2d** features an aliphatic side chain, it also facilitated selective peptide bond formation with R, albeit at a lower level than **2b**. Overall, these findings emphasize that both side chain interactions and self-assembly can work together to slow down reactivity while enhancing selectivity in peptide coupling. The UPLC-MS chromatograms and peptide bond conversions for the systems containing **2b** and amino acid mixtures are available in the Supplementary Information (Supplementary Figs. 79–87). Additional TEM images for the systems involving **2a** and **2b** are provided in the Supplementary Information (Supplementary Figs. 88–90).

### Effect of phosphate ester on peptide coupling

Following our findings on the role of the side chains ($R^2$ in Fig. 1B) in driving selective peptide bond formation from mixtures of natural and non-natural amino acids, we then explored the effect of the phosphate ester ($R^3$ in Fig. 1B). Thus, we synthesized a derivative by retaining the Fmoc group on the *para* position of the phenylalanine residue while substituting the ethyl with a phenyl ester (**Boc-F(4-NH-Fmoc)PP**)— referred to as molecule (**3**). This modification aimed to test whether aromatic phosphate esters could exert additional selection pressure by utilizing both aromatic stacking and electrostatic interactions from the charged phosphates (Fig. 3A). In the first set of experiments, building block **2b** was mixed with a combination of R and the non-natural amino acid F(4-guanidine), where a preference for R in peptide bond formation was observed. However, this preference reversed when using building block **3**, suggesting that the aromatic phenyl ester in **3** interacted more favorably with F(4-guanidine) (Supplementary Figs. 91–94). TEM images of the mixture displayed a similar assembly

behavior to that seen with R alone, reinforcing the higher peptide bond conversion for R. In contrast, this distinct assembly behavior was absent when **3** was used (Supplementary Fig. 95). In the subsequent individual library experiments, the phenyl ester-containing aminoacyl phosphate (**3**) showed a significantly higher yield of peptide bond formation with both F and F(4-guanidine) compared to **2b** (Fig. 3B, C). These results suggest that aromatic stacking interactions in **3** play an important role in enhancing efficiency for peptide bond formation compared to **2b**.

Having established the effect of single amino acids on peptide bond formation, we next investigated how dipeptide sequences could influence this process. The motivation for these experiments was to evaluate whether peptide substrates, could impact selectivity and overcome the limitations observed with negatively charged amino acids like aspartic acid. We started by examining the dyad of aspartic acid (DD), which, as expected, exhibited only trace amounts of peptide bond formation with both **2b** and **3** (Supplementary Figs. 96 and 97). However, replacing the C-terminal aspartic acid with arginine (DR) and tryptophan (DW) significantly increased peptide coupling yields (Supplementary Figs. 98–101). We further explored mixtures containing two dipeptide sequences, including DR, DW, and DD, DW. In these mixtures, **2b** (ethyl phosphate ester) primarily led to hydrolysis as the dominant product. In contrast, **3** (phenyl phosphate ester) favored the formation of peptide bond with DW as the main product (Fig. 3D, E, and F). This trend remained consistent even in mixtures containing three dipeptide sequences (DD, DR, and DW) (Supplementary Figs. 102–106). These results suggest that the aromatic nature of the phenyl phosphate ester in **3** enhances selectivity, particularly with aromatic residues like tryptophan (W). To further confirm this selectivity, we tested another mixture containing DD and DF with **2b** and **3**. The results revealed a similar behavior, where DF showed selective coupling with **3**, while **2b** predominantly resulted in hydrolysis (Supplementary Figs. 107–109). Notably, the high peptide bond conversion observed with DW suggests that the indole ring of tryptophan enhances the efficiency of peptide coupling and contributes to the stabilization of supramolecular assemblies[56]. Cryo-TEM imaging revealed that the phenyl phosphate ester (**3**) assembled into spherical aggregates, distinguishing its supramolecular behavior from **2b**, which formed smaller assemblies (Fig. 3G). DLS analysis further confirmed that the aggregates formed by **3** were slightly larger than those of **2b** (Supplementary Fig. 110). Interestingly, despite hydrolysis producing the same carboxylic acid derivative for both esters, time-dependent microscopy revealed distinct fibrillar organization between the two, suggesting that interactions between the phenyl phosphate ester and the carboxylate derivative influenced the resulting assembly (Fig. 3G and Supplementary Figs. 111–113). The dynamic changes in the aggregation process of **3** were further monitored using fluorescence spectroscopy, which revealed a gradual decrease in intensity, indicating reorganization of the aromatics (Supplementary Fig. 114).

### Covalent self-sorting versus co-assembly

Building on our previous observations with N-terminus protected aminoacyl phosphate esters, we next explored peptide oligomerization using N-terminus free aminoacyl phosphate esters. We hypothesized that assembly of aminoacyl phosphate esters might allow for controlled oligomerization (Fig. 4A). Thus, we synthesized derivatives incorporating aromatic amino acid residues: FEP (**1a**), F(4-NH-Fmoc)EP (**1b**), VEP(**1d**), F(4-guanidine)EP (**1e**), BPAEP (**1f**, where BPA stands for 4-benzoyl-phenylalanine) and F(4-COOH)EP (**1g**). These activated amino acids were synthesized as ethyl esters. When **1b** and **1f** were mixed at equimolar concentrations (10 mM each) in 0.6 M borate buffer (pH 9.1), we observed the formation of mixed oligomeric species (Fig. 4B, Supplementary Figs. 115–117 and Supplementary Table 2). Microscopy studies revealed that both **1b** and **1f** formed spherical aggregates, similar to what was observed with previously studied protected

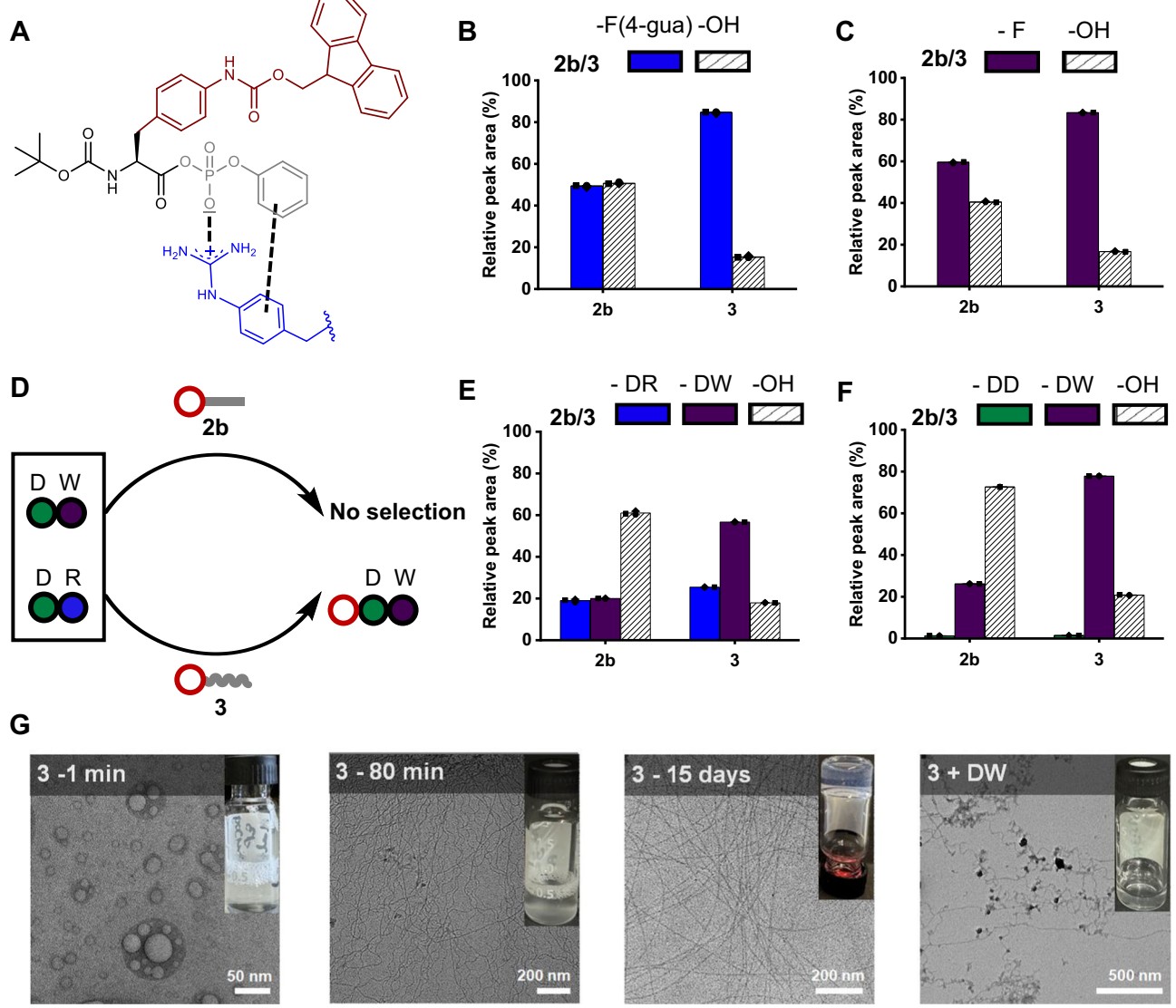

**Fig. 3 | Effect of phosphate ester on peptide coupling. A** Schematic representation of the interactions involved between aminoacyl phosphate ester 3 and a phenylalanine amino acid residue with a guanidine group in the *para* position. Peptide bond formation between **2b** or **3** with **B** 10 mM F(4-guanidine) and **C** 10 mM F. The concentration used for **2b** or **3** was 10 mM and the samples were prepared in 0.6 M borate buffer pH 9.1. Peptide coupling was measured after 48 h. **D** Schematic representation of a competition experiment between **2b** or **3** with a dipeptide mixture containing DR and DW. Peptide bond formation between **2b** or **3** with mixture of dipeptides containing **E** DR and DW (10 mM each) and **F** DD and DW (10 mM each) in 0.6 M borate buffer pH 9.1. The concentration used for **2b** or **3** was 10 mM. In the bar graphs, **2b** and **3** correspond to ethyl and phenyl phosphate esters, respectively. The colors of the bars represent the products formed when **2b** or **3** are mixed with individual amino acids and dipeptides. Striped bars indicate hydrolysis products (**3-OH** or **2b-OH**). Error bars represent the standard deviation of three independent experiments. Peptide coupling yields were measured after 72 h. **G** Cryo-transmission electron microscopy images (from left to right) of 10 mM **3** immediately after dissolving, after 80 min, after complete hydrolysis (15 days), and upon coupling with 10 mM DW.

aminoacyl phosphate esters like **2b** and **3** (Supplementary Figs. 118 and 119). Although no hydrogel formation was observed in the BPA-containing libraries, a weak hydrogel formed in the Fmoc-containing samples after oligomerization. Rheological studies further suggested that the incorporation of mixed sequences enhanced the mechanical properties of the resulting structures, pointing towards a co-assembly effect between oligomers (Supplementary Figs. 120 and 121). The role of guanidine residues was also important in these systems. When guanidine was present (**1e**), it promoted the formation of hetero-oligomers through electrostatic interactions with the phosphate esters, highlighting its importance in influencing the construction of mixed species (Supplementary Figs. 122–125 and Supplementary Table 3).

Next, we explored the behavior of libraries containing **1b** and **1g**. This mixture resulted in two distinct families of oligomers enriched in either Fmoc- or F(4-COOH)-containing species, indicating covalent self-sorting (Fig. 4C, Supplementary Figs. 126–128 and Supplementary Table 4). Centrifugation experiments revealed distinct phases, where the aggregated phase was predominantly enriched with **1b** oligomers, while **1g** oligomers remained in solution (Supplementary Fig. 129). Confocal microscopy further confirmed that **1b** formed spherical aggregates that transitioned into fibrillar structures over time, whereas **1g** did not show aggregate formation (Supplementary Fig. 130). Fluorescence spectroscopy also supported these findings, as the emission spectra of mixed solutions containing **1b** and **1g** showed no change compared to **1b** alone, indicating that phase separation prevented interaction between the species (Supplementary Fig. 131). Self-sorting was also observed in libraries containing **1b** and **1a** (Supplementary Figs. 132–135 and Supplementary Table 5). To confirm the generality of this process, we extended the system to aliphatic

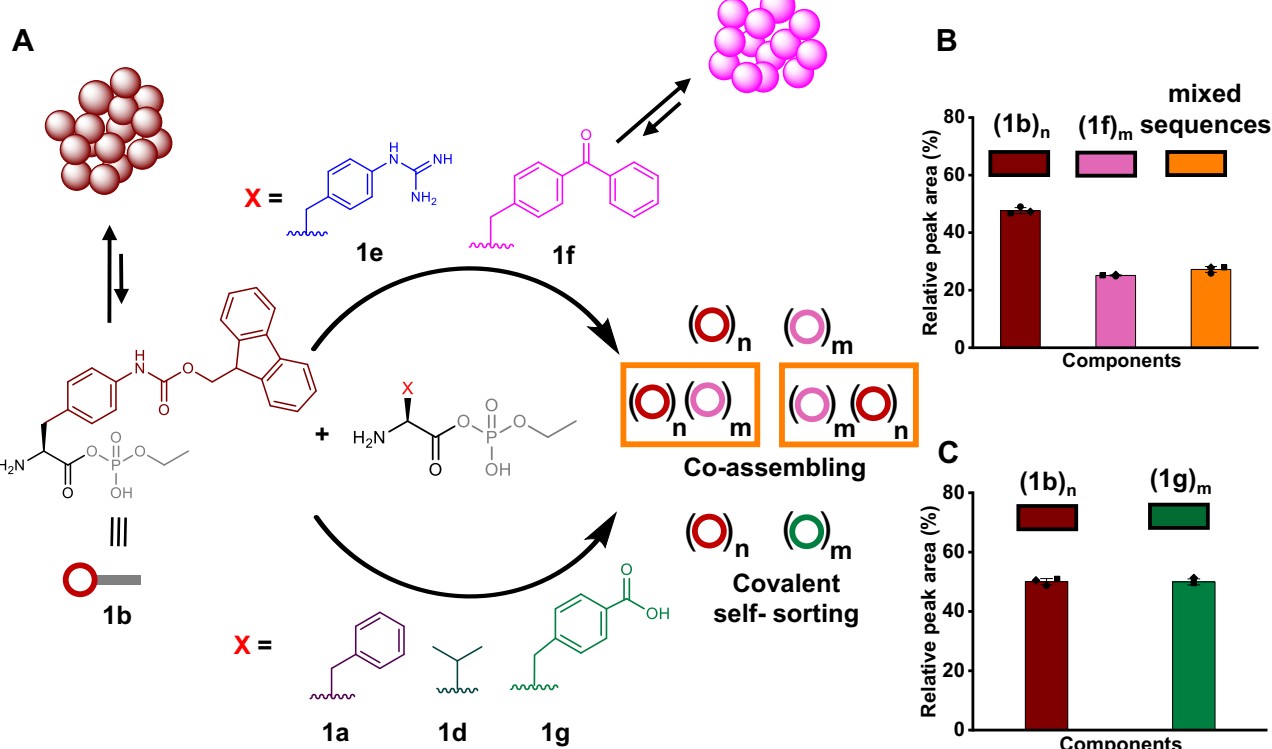

**Fig. 4 | Covalent self-sorting and co-assembly in spontaneous oligomerization reactions. A** Schematic representation of the oligomerization reaction between **1b** and other amino acyl phosphate esters (**1a, 1d, 1e, 1f,** and **1g**) resulting in covalent self-sorting or co-assembly. Formation of homo and hetero-oligomers between 10 mM **1b** with **B** 10 mM **1f** and **C** 10 mM **1g**. Libraries were prepared in 0.6 M borate buffer, pH 9.1. Peptide coupling was measured after 24 h. Bar graphs represent the sum of all library components (grouped as homo or hetero-oligomers) formed in the reactions. Error bars represent the standard deviation of three independent experiments.

aminoacyl phosphate esters, VEP (**1d**) (Supplementary Figs. 136–139 and Supplementary Table 6). Overall, these findings suggest that the assembly of aminoacyl phosphate esters creates a protective microenvironment that positions primary amines in close proximity to the acyl phosphate esters, facilitating selective coupling and homo-oligomerization. However, when additional non-covalent interactions, such as aromatic stacking (as seen with **1f**) or electrostatic forces (as observed with **1e**), are introduced, the aggregates engage in co-assembly, leading to the formation of hetero-oligomers. This balance between covalent self-sorting and co-assembly offers a versatile strategy for controlling peptide oligomerization. It allows specific amino acid residues to interact preferentially within different phases (aggregated vs. solution). As a result, the process drives the selective formation of oligomeric species with distinct chemical compositions.

## Discussion

Our study demonstrates that selective peptide bond formation in aqueous environment can be controlled by adjusting the structural features of aminoacyl phosphate esters. We show that these esters can be tuned to promote specific interactions between their phosphate ester groups and amino acid side chains, creating a versatile platform for achieving selectivity in peptide bond formation. Systematic modifications of the side chains, from aliphatic to aromatic, revealed distinct reactivity patterns, with self-assembly playing a key role in slowing reaction rates and enhancing selectivity. We demonstrated that coupling can be tailored to the structure of specific polar phosphate esters. While coupling with positively charged amino acids was generally preferred, the transition from ethyl to phenyl phosphate esters significantly improved coupling with aromatic residues, highlighting the role of non-covalent interactions in driving peptide bond formation. Interestingly, the limited reactivity of negatively charged

amino acids was overcome by incorporating them into dipeptide sequences. This highlights that the exact primary sequence is less critical for coupling, as the phosphate esters and amino acid side chains work together to promote specific interactions, enabling the formation of otherwise challenging sequences. Our approach utilizes these polar head groups to create an environment that enables the incorporation of amino acids that typically resist coupling, thus expanding the range of accessible peptide sequences driven by aminoacyl phosphate esters. The interactions between amino acid side chains and phosphate ester groups also drive the spontaneous formation of diverse supramolecular structures, such as fibers, micelles, and ribbons, providing a flexible platform for generating tunable peptide-based materials. Additionally, we extended this strategy to peptide oligomerization, where N-terminus-free aminoacyl phosphate esters capable of self-assembly facilitated covalent self-sorting or co-assembly based on the hydrophobicity of the amino acids involved. These findings highlight the potential of self-assembly to drive selective peptide bond formation in a bioinspired approach, independent of complex biological machinery. In the future, we aim to apply this strategy to sequential peptide couplings in solution, with the goal of achieving selectivity and orthogonality in the formation of longer peptide oligomers.

## Methods
### Materials

All reagents were purchased from Sigma-Aldrich and Carl Roth and used without any further purification unless otherwise indicated. Boc protected and N-terminus free amino acids were purchased from Carbolution, ABCR, and VWR. Aluminium metal plates precoated with silica gel 60 matrix, 0.25 mm or 0.5 mm were utilized for thin-layer chromatography (TLC). Visualization of the developed TLC plate was

performed by irradiation with UV light. Preparative RP-MPLC was performed using an automated Interchim-puriFlash® system.

**Synthesis of aminoacyl phosphate esters.** The detailed synthesis of the activated amino acids is provided in the Supplementary Information of the manuscript.

**Library preparation.** Amino acids were dissolved in 0.6 M borate buffer (pH 9.1) and 0.1 M PBS buffer (pH 7.5 and 8.1). The freshly prepared amino acid solutions were then transferred to a new vial containing the aminoacyl phosphate ester, followed by vortexing and sonication. The concentration of aminoacyl phosphate esters (either protected or N-terminus free) used in all libraries was 10 mM.

**UPLC analysis.** UPLC analyses were performed on a Waters Acquity UPLC H-Class Biosystem, equipped with a photodiode array detector at a detection wavelength of 214 nm. Samples were injected on an Acquity UPLC CSH-C18 ($150 \times 2.1$ mm) column, using ULC-MS grade water eluent (A) and ULC-MS grade ACN eluent (B), which contained 0.1% trifluoroacetic acid as the modifier. A flow rate of 0.3 ml min$^{-1}$ and a column temperature of 35 °C were applied. For all samples, vortex (30 s) and sonication (30 s) were performed prior to UPLC injection to ensure a homogeneous phase. Vortex and sonication (for 15 s) were furthermore applied after dilution, prior to UPLC injection. Samples were prepared by taking 10 μl from the reaction vial and diluting (100 times) into $H_2O$ or $H_2O$: ACN or $H_2O$: THF mixture. The solvent ratios were selected based on the hydrophobicity of the library components. UPLC samples consisting of **2a** were prepared in 100% water, while samples of **2b** were dissolved in a 30% ACN: $H_2O$ mixture. Samples containing mixed aminoacyl phosphate esters (**1b** with **1a**, **1c**, **1d**, **1e**) were dissolved in a 30% THF: $H_2O$ mixture.

**UPLC-MS analysis.** UPLC-MS experiments were performed on an Agilent 6546 LC/Q-TOT equipped with an infinity 1290 II in the LC section. We used the same UPLC column, as this is described in the UPLC analysis section above. The Q-TOF was equipped with a dual AJS ESI source. The experiments were conducted at a VCap voltage of 4000 V, a sheath gas temperature of 300 °C and a fragmentor voltage of 120 V. An internal reference was used.

**Peptide coupling in libraries containing protected aminoacyl phosphate esters.** With the relative peak area (%) of each component, we refer to the percentage of peptide coupling formed from amino acids and dipeptide sequences.

**Peptide coupling in libraries containing N-terminus free aminoacyl phosphate esters.** We refer to the peak area (%) of single amino acids and oligomers relative to the absorbance of the monomeric unit. In Fig. 4B, C, in libraries where two aminoacyl phosphate esters were mixed to produce various combinations of oligomers, the total peptide coupling is calculated as the sum of all the respective components formed in the system. Since the peak areas of monomers differ, a correction factor (X) is applied, by dividing the area of the monomer with greater value by the area of the monomer with lesser area. For example, in case of the library made from **1b** with **1e**, we adjusted the area as follows:

1. Correction Factor (X):
   X = (Area of monomer **1b**)/(Area of monomer **1e**)
2. Adjusting the area of monomer **1e**:
   Adjusted Area of monomer **1e** = X * (Area of monomer **1e**)

**NMR.** The $^1$H NMR spectra were recorded on a Bruker Avance Neo 400 and 300 MHz with broadband cryoprobe Prodigy. The $^{31}$P NMR spectra were recorded on 162 and 122 MHz spectrometers using $^1$H-broad band decoupling in the indicated deuterated solvent. Chemical shifts were reported as delta values from standard peaks.

**Rheology.** Rheological measurements were carried out with an Anton Paar MCR 302 rheometer at 20 °C, using a 25 mm cone-plate geometry (CP25-1, Anton Paar) and a measuring gap of 0.047 mm. Samples were prepared as previously mentioned and placed on the bottom plate.

**Transmission electron microscopy (TEM).** Libraries were vortexed and sonicated prior to sample preparation for imaging. Small drops of the solution, the gel, or the suspension were applied to a carbon-coated Cu grid for 30 s incubation, followed by two drops of water wash and one drop 5 μl of 2% (w/v) uranyl acetate solution for 30 s staining. Excess solution was removed by blotting the grid with a piece of filter paper and left to air dry. Imaging was performed using a FEI Talos 120C at 120 kV operating voltage. Images were taken using a Ceta 16 mega pixel camera.

**Cryo-transmission electron microscopy (Cryo-TEM).** Cryo-EM was used to visualize the morphology of samples **2b**, **3**, **1b**, and **1d** in their native state. A C-flat holey carbon grid with standard 20 nm carbon thickness, 2.0 μm hole diameter, and 300 mesh Cu grid (CF-2/4-3Cu from Protochips Inc., North Carolina, USA) was hydrophilized by 2 min of glow discharge using air and a remote source at 15 W in a Tergeo EM Plasma Cleaner (PIE Scientific LLC, California, USA). Sample at 10 mM concentration, was first dissolved in the 0.6 M borate buffer, pH 9.1, and then applied to the carbon grid at 22 °C. The sample was vitrified by automated blotting and plunge freezing with an FEI Vitrobot Mark IV (Thermo Fisher Scientific Inc.) using liquid ethane as the cryogen. Cryo-EM image acquisition: The vitrified specimen was transferred to an FEI TALOS L120C electron microscope (Thermo Fisher Scientific Inc.) with an acceleration voltage of 120 kV using a 626 single tilt liquid nitrogen cryo-transfer holder (Gatan Inc.). Images were captured using a CETA camera.

**Dynamic light scattering (DLS).** DLS measurements were recorded using a Malvern Zetasizer Nano ZSP instrument. Samples of **2b** and **3** at different concentrations, were dissolved in 0.6 M borate buffer, pH 9.1, and were immediately measured at 25 °C.

**Fluorescence spectroscopy.** Solutions of aminoacyl phosphate esters (1.5 ml) were prepared in 0.6 M borate buffer, pH 9.1, and pipetted into a quartz glass cuvette of $10 \times 10$ mm light path (Hellma analytics). Emission spectra were recorded at different time points in Jasco FP-8300. The emission spectra were measured between 305 and 700 nm. Excitation was at 295 nm at 25 °C and a scan speed of 500 nm per minute.

**Confocal microscopy.** Imaging was performed using a Zeiss LSM 710 confocal microscope equipped with a 63x oil immersion objective lens. Samples were prepared as previously described and transferred into micro-well plates (ibidi, μ-Slide 8 well Bioinert) for the imaging. Nile red (1 μM) was added as a fluorescent dye. Samples were excited at 561 nm and imaged within the 575–630 nm range.

## Data availability
All data generated or analyzed during this study are included in this article and its supplementary information files, and are available from the corresponding author. Source data are provided with this paper.

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

## Acknowledgements

This work was supported by Deutsche Forschungsgemeinschaft (DFG, German Research Foundation) under the project (PA 3783/2-1) and the European Union (ERC-2023-StG grant, PhosphotoSupraChem, 101117240). We thank Dr. Victor Hugo Pacheco Torres from the Institute of Macromolecular Chemistry at the University of Freiburg for the support with measuring NMR samples. We thank Christoph Warth from the Institute of Organic Chemistry at the University of Freiburg for the analytical support.

## Author contributions

C.G.P. conceived the idea and supervised the overall project. A.S. performed the synthesis of the aminoacyl phosphate esters and the analysis of the libraries using UPLC-MS. K.D. performed the rheology experiments and assisted with the analysis of the libraries using UPLC-MS. M. P. assisted with the synthesis and purification of the activated amino acids. R.T. and Y.T. performed all the TEM and cryo-EM experiments. S.K.R. assisted with the NMR analysis. C.G.P. and A.S. co-wrote the manuscript with the support of all authors.

## Funding

## Competing interests

The authors declare no competing interests.
