## [Transparent Peer Review file · Nature Communications]

Selective Peptide Bond Formation via Side Chain Reactivity and Self-Assembly of Abiotic Phosphates

Corresponding Author: Dr Charalampos Pappas

Version 0:

Reviewer comments:

Reviewer #1

(Remarks to the Author)

Navigating beyond randomness....

The paper describes a new strategy for the selective incorporation of (positively charged) amino acids from mixtures containing natural and non-natural amino acids. This is achieved by the design of aminoacyl phosphate esters that can assemble into supramolecular aggregates (not very well characterized or defined) and react rapidly with positively charged (Arg, Lys) aas versus different non-charged aas.

Unfortunately, the paper is so badly written – with respect to experimental description, coherent writing, data analysis and data discussion– that I didn't bother reading beyond page 7 (up to figure 2). The many issues described below (not a complete list) made it almost impossible to assess the validity of the data; my recommendation is to reject the paper. If the authors wish to submit it somewhere else, they would like to go over the entire manuscript and upgrade it in many aspects.

Examples of issues arising from the first sections of the paper (this is not a complete review of the manuscript).

p. 5 line 137-8. "...we determined high yields of peptide bond formation (70-80%)...". After how much time? Is this a kinetic resolution or a result obtained at the end of the reaction?

p. 5 line 138. "...indicating that the pKa and the chemical structures of amino acid side chains have minimal impact on coupling". This is not a clear statement. The alpha-amino protons of all amino acids have more or less the same pK....

p. 5 line 139-40. "This observation is particularly evident in the utilization of secondary amines, such as proline (P)." But proline was not tested in mixtures I or II. What is the value of this comment? Then "Despite challenges associated with peptide coupling when proline residues are used, we achieved up to 85% conversion in samples containing 2a and P." But where is the data – we don't see it in the main manuscript nor in the SI.

p. 5 line 144-45. The text describes experiments with 1:1 and 1:5 ratios... but it doesn't say anywhere the ratio between what. By more in depth reading into the SI figures, one finds that the ratio of 1:5 is between the aminoacyl phosphate ester and the total amount of aas in the studied mixture. No reasoning, hypothesis or implications are provided for the two different experiments and their results.

Fig. 2 c,d,e,f,g and related text at page 5. At what time or reaction stage were the relative product ratios determined?

p. 5 line 161. "Notably, the system maintained compartment formation, but larger aggregates were visualized when coupled with R...". Coupled or mixed?

Bottom of p. 5 and beginning of p. 6. The system discussion 'navigates' freely between mixtures containing R or K. There is no explanation for that... and it is not always clear when one ends and the other starts.

p. 6 line 187 and on. "...perform experiments with all 20 natural amino acids in one pot. Both for 2a and 2b, we observed that peptide coupling with cysteine (C) via thioester bond formation predominated over other amino acids (Fig. 2H)." This result is completely orthogonal to the paper's study. It is perfectly clear why Cys reacted faster... Since Cys reacts via NCL which is both efficient and selective, the analysis with Cys comprises a completely different aspect. The other aspects of the study with all 20 aas are described only in the SI. This is weird because they provide the most essential and interesting data.

Incoherent/unjustified text and terminology

- P. 4 line 100. "integrate structural elements in their structure for preorganization into compartments". Instead of 'amphiphilic'.

- P. 4 line 104-5. "... where the phosphate esters act as solubility tags..". Instead of charged/polar residues increasing

solubility.

- P. 5 line 127. At the end of the introduction the manuscript mentions: "We also attempted experiments with all twenty natural amino acids in one pot, with the aim of determining the effect of amino acids with nucleophilic side chains on peptide bond formation." But, what did you get here? This is also not so clear when reading the results sections (see comments above).

One can go on and on here.

Reviewer #2

(Remarks to the Author)

This work reported an amide bond formation strategy using aminoacyl phosphate esters. By tuning the phosphate ester structure, the peptide bond could be selectively formed according to the molecule electrostatic charge or aromaticity. Besides, processes including self-sorting and co-assembly could be controlled. This strategy is promising in sequential peptide coupling with selectivity and orthogonality. However, there are several issues requiring further elaboration before this work can be recommended for publication on Nature Communications.

1. The phrases "preorganization" and "preorganization into compartment" are quite confusing, since there is no extra process described before reaction with amino acids in the solution. Do they mean structural modification or structural regulation? Further elaboration on this issue is necessary. Exchange to easier-understanding words would be better.

2. The selectivity, or the departure ability from randomness, of the phosphate ester is suggested to be quantitatively determined by the thermodynamic reaction free energy, which is not hard to achieve. This would be more explicit and persuasive than the simple ratio percentages (Lines 141, 146, and 147).

3. Most of the aminoacyl phosphate esters designed in this work are non-natural amino acid derivatives. It is interesting how is the feasibility to use natural amino acids as the phosphate ester. Additional discussion on this issue is recommended.

4. According to Figure 3C, the selective production of -R and -F(4-gua) is not remarkable, so that the schematic Figure 3B is somehow misleading. The Figure 3E and Figures 3FG are also contradict, because 3E describes a three-component mixture while 3FG compare only 2 fractions. Revision on these contents is requested. By the way, since the pairs of EP/PP and 2b/3 have the same structural difference, could EP/PP be exchanged to 2b/3 for clarity?

5. Covalent self-sorting of 1b and 1e without any production of mixed sequence is amazing. A deeper analysis on the mechanism in addition to the hydrophobicity/hydrophilicity is highly encouraged.

6. The English written requires further polishing.

Minor issues:

7. In Figure 1A, the R1 and R2 group of compound 3 has no definition.

8. In Figure 2A, the R group of compound 2a/2b-X should be X.

Reviewer #3

(Remarks to the Author)

In this manuscript, Sharma et al. discuss how chemical modifications designed on aminoacyl phosphate esters affect peptide synthesis. The authors observe that specific activated substrates result in supramolecular structures that affect product distribution. The manuscript contains exciting results, including the control over the formation of hetero or homo-oligomers upon the design of the reactants. However, I found the manuscript not well-balanced: a too-long introduction that doesn't go deep enough on what's the state of the art in selective peptide synthesis in water, a very superficial characterization of the supramolecular assemblies, and no effort to explain the effect of different supramolecular assemblies on the results; a discussion that claims selective peptide formation when the reactions reported are not as selective as the authors claim; a final result (long oligomers synthesis) that would have deserved more than half a page and a more thorough explanation and characterization.

More in detail:

Figure 1 is confusing, as it comes too early in the manuscript when the readers haven't yet read about all the results. There are too many catchy words, with not enough context because everything will be explained much later in the text. I suggest moving this figure to the end, as it helps summarize the results.

In the experiments with all 20 amino acids, the predominant product is always the Cy-containing peptide. Even though the authors show they can selectively control the formation of the other products, they do not have any control over the main product. This result seems to undermine the whole point of the manuscript while not adding much. I would consider removing the set of experiments.

- Tables in the SI (for example, Tables 2 and 3) would be more informative if yields/product distribution were included. Also, yields are rarely given in the main text, leaving the readers to guess what worked better by comparing the height of bars in the graphs.

- The authors report the selection for peptide bonds formed with R and K within mixtures. R and K are rather different positively charged amino acids, including the ability of R, but not of K, to interact with aromatics. The readers are thus left wondering if there is any preference or competition between R and K when they are both present as reactants. The manuscript would gain in depth if the authors would perform this experiment (and explain the results).

- Additionally, the authors cite Powner's work on the selective synthesis of Lys-containing peptides. Have the authors tested ornithine and diaminopropionic acid in place of lysine? What results would the authors expect?

- Figure 2, panel E: why is F more efficient at forming peptide bonds in ACN compared to L? And why is F so efficient in the presence of sodium sulfate?
- It would be helpful if the authors could state more clearly in the main text when the samples were analyzed by UPLC, considering that the aggregates' morphologies change over time. Does the product distribution change if the sample is analyzed after 10 minutes, 1 hour or 10 days?
- Many TEM and CryoEM images in the paper are confusing. At times, there are dark black spheres, at times, droplets, and at times, fibers. Are these samples amorphous or (partly) crystalline? The authors report DLS data that point towards tiny aggregates (10 nm), but the size of the aggregates observed by EM is very different. Overall, the characterization of the aggregates is very superficial, yet the authors try to extrapolate information about the number of layers present in the aggregates. Light microscopy would have probably been a better choice for the characterization of the aggregates. The authors should also explain the inconsistencies in the size of the aggregates. Finally, the authors make no effort to hypothesize why aggregates (and which ones) control the selectivity of peptide bond formation. For example, is the aggregate of the aminoacyl phosphate ester the one that controls the selectivity, or is the co-assembly of the aminoacyl phosphate ester with the amino acid (like R) that controls it?
- In the dipeptide experiments, why did the authors select tryptophan? It seems odd, considering they haven't tested W before. Why not use DF if the authors aim to test the effect of aromatics? Also, why did the authors change the amino acid at the C-term instead of at the N-term? Does the order have any influence?
- Figure 3, panel A: I found it confusing that the authors highlight the electrostatic interaction between Arg and the phosphate (on the left side). This same interaction should also occur in 2a, where R2 is different. This interaction does not explain why no selectivity is observed in 2a.
- Figure 3, panel H: It is unclear why the authors define these aggregates of droplets/bubbles as multilayer. I suggest removing it because it is not apparent in the EM images.
- As mentioned, I found the last paragraph the most interesting of the manuscript, yet it is very superficial and not discussed. The covalent-self-sorting results are really clean and, if I understood well, suggest that 1e does not "partition" in the aggregate with 1b, so polymerization occurs in the solution. More explanation as to why this happens and how general it could be would be needed.

Version 1:

Reviewer comments:

Reviewer #2

(Remarks to the Author)

This revised manuscript basically addressed my previous questions. The addition greatly improved the quality of this work. It can be thus recommended for publication at this stage.

Reviewer #3

(Remarks to the Author)

The authors have done an incredible job at rewriting most of the manuscript, which now reads very well. The additional experiments I suggested have been conducted, and the explanations I requested have been addressed and included in the main text. Figures are now clear and descriptive. I believe this manuscript is now suitable for publication in Nature Communications. I am also convinced it will be of extreme interest and a source of inspiration for the systems, biomimetic, and supramolecular chemistry communities.

Editorial note: Reviewer 3 has assessed authors responses to points of Reviewer 1 and considers them fully addressed.

Reviewer: 1

Navigating beyond randomness....

The paper describes a new strategy for the selective incorporation of (positively charged) amino acids from mixtures containing natural and non-natural amino acids. This is achieved by the design of aminoacyl phosphate esters that can assemble into supramolecular aggregates (not very well characterized or defined) and react rapidly with positively charged (Arg, Lys) aas versus different non-charged aas. Unfortunately, the paper is so badly written – with respect to experimental description, coherent writing, data analysis and data discussion– that I didn't bother reading beyond page 7 (up to figure 2). The many issues described below (not a complete list) made it almost impossible to assess the validity of the data; my recommendation is to reject the paper. If the authors wish to submit it somewhere else, they would like to go over the entire manuscript and upgrade it in many aspects.

Response: We thank the reviewer for his/her feedback and recognize the importance of presenting the data and discussion in a clearer manner. In response, we have made substantial revisions throughout the manuscript to improve experimental descriptions, data analysis and the overall clarity of the writing. Specifically, we have addressed the characterization of supramolecular aggregates by conducting additional experiments, including fluorescence spectroscopy and microscopy, as well as DLS, which are now thoroughly detailed in the revised version. We have simplified the figures and improved the clarity of the results and discussions. Additional context and details have been provided to ensure the validity of the data can be easily assessed.

While we regret that the reviewer did not read beyond page 7, we would like to emphasize that much of the manuscript's most compelling data, including the synthesis tailored to phosphate esters and self-sorting experiments, appears later in the paper. These experiments offer crucial insights into how selective peptide coupling is driven by side chain reactivity and self-assembly of abiotic phosphate esters. The results clearly demonstrate the differentiation between positively charged and aromatic amino acids in complex mixtures, as well as the role of supramolecular aggregates in enhancing reaction selectivity. Furthermore, the data highlight how phosphate ester reactivity can be fine-tuned to induce sorting behavior, with important implications for systems chemistry and chemical reaction networks. These findings advance our understanding of selective peptide coupling and self-assembly, and we believe they are highly relevant to ongoing research in peptide synthesis and dynamic self-assembly. We hope that the revised manuscript, with improved clarity, will allow these novel results to be more easily appreciated.

Examples of issues arising from the first sections of the paper (this is not a complete review of the manuscript).

p. 5 line 137-8. "...we determined high yields of peptide bond formation (70-80%)...". After how much time? Is this a kinetic resolution or a result obtained at the end of the reaction?

Response: The yields of peptide bond formation were determined after the complete hydrolysis of the phosphate esters, as confirmed by the absence of remaining phosphate ester peaks in the UPLC chromatograms. Specifically, the reactions (referred in line 137-138) were monitored and completed after 60 minutes. This clarification regarding the reaction time of all systems has been added to the manuscript in all relevant sections.

p. 5 line 138. "...indicating that the pKa and the chemical structures of amino acid side chains have minimal impact on coupling". This is not a clear statement. The alpha-amino protons of all amino acids have more or less the same pK....

Response: We agree that the statement could be clearer. The intent was to convey that the differences in the pKa values of the α -amino groups of the amino acids do not significantly affect the efficiency of the coupling reaction under our experimental conditions. The coupling yields remained high across

amino acids with different side chains and pKa, indicating minimal impact on the peptide bond formation process.

We revised the following text in the manuscript: “Next we explored peptide bond formation using **2a** with various amino acids in individual reactions, such as D, S, L, R, F, K, H, W, P and I. Peptide bond formation reached 70-80% for most tested amino acids, as confirmed by ultraperformance liquid chromatography mass spectrometry (UPLC-MS), indicating a highly efficient and spontaneous process under these conditions.^{53,54} These results show that slight differences in the pKa of the α -amino groups of the amino acids had minimal effect on the efficiency of the coupling reaction.”

p. 5 line 139-40. “This observation is particularly evident in the utilization of secondary amines, such as proline (P).” But proline was not tested in mixtures I or II. What is the value of this comment? Then “Despite challenges associated with peptide coupling when proline residues are used, we achieved up to 85% conversion in samples containing **2a** and P.” But where is the data – we don’t see it in the main manuscript nor in the SI.

Response: Proline was included in the study as a representative secondary amine to investigate the extent of peptide coupling separately. We achieved up to 85% conversion with **2a** and proline, whereas the conversion with **2b** was 58%. Following the reviewer’s suggestion, we conducted additional experiments by incorporating proline into mixture I, replacing aspartic acid, since the coupling with aspartic acid was minimal. The results of these experiments have been added to the Supplementary Information.

We revised the following text in the manuscript: “Further examining the impact of different side chain configurations, we introduced proline (P) in the original amino acid mixture I. However, this modification did not alter the coupling behavior in either the **2a** or **2b** containing libraries (Supplementary Fig. 56-60).”

Supplementary Figure 60: Bar graphs showing peptide coupling between A) 10 mM **2a** and B) 10 mM **2b** with 50 mM amino acid mixture (P, S, R, F, L, each amino acid is 10 mM), in 0.6 M borate buffer, pH 9.1. In each bar graph striped bar represent the hydrolysis product **2a-OH/2b-OH**. Peptide coupling yields were measured for **2a** after 30 minutes, while for **2b**, after 48 hours.

p. 5 line 144-45. The text describes experiments with 1:1 and 1:5 ratios... but it doesn't say anywhere the ratio between what. By more in depth reading into the SI figures, one finds that the ratio of 1:5 is between the aminoacyl phosphate ester and the total amount of aas in the studied mixture. No reasoning, hypothesis or implications are provided for the two different experiments and their results.

Response: We appreciate the reviewer's comment and have clarified that the 1:1 and 1:5 ratios refer to the molar ratio between the aminoacyl phosphate ester and the total amount of amino acids in the mixture. The purpose of varying these ratios was to explore their effect on hydrolysis and peptide coupling. As demonstrated in our previous reports,^{1,2} hydrolysis is favored when stoichiometric amounts of the reactants are used. By adjusting the ratio, we aimed to shift the balance in favor of peptide coupling and mitigate hydrolysis, optimizing the formation of coupling products over hydrolysis byproducts.

We have now added the following text in the manuscript: "In order to mitigate hydrolysis and promote peptide bond formation, we increased the ratio of aminoacyl phosphate ester to amino acid mixture I to 1:5. In these experiments, we observed a slight increase in peptide coupling with R, reaching 35%. In the same mixture, F showed a coupling efficiency of 23%, while S reached almost 20% (Fig. 2C). There were no significant changes observed when libraries prepared at different pH (Supplementary Fig 2B and 2C), suggesting that product distribution was mainly influenced by the intrinsic reactivity of the phosphate ester. The UPLC-MS chromatograms and peptide bond conversions for the systems containing 2a and amino acid mixtures are available in Supplementary Information (Supplementary Figs. 3-21)."

Fig. 2 c,d,e,f,g and related text at page 5. At what time or reaction stage were the relative product ratios determined?

Response: We have now added the reaction times for all systems in both the main text and the relevant figure legends.

p. 5 line 161. "Notably, the system maintained compartment formation, but larger aggregates were visualized when coupled with R...". Coupled or mixed?

Response: The aminoacyl phosphate ester was first mixed with amino acid mixture I, which included arginine, leading to selective peptide bond formation where the arginine-containing product was predominant. The subsequent assembly process was visualized using TEM, Confocal Microscopy, and Fluorescence Emission Spectroscopy after peptide bond formation. In particular, monitoring a library containing 10 mM 2b with 10 mM arginine (R) via Confocal Microscopy over time revealed a transition from smaller aggregates to larger ones, indicating that the peptide coupling product of R with 2b drove aggregation. This was further supported by Fluorescence Emission Spectroscopy, where a significant decrease in intensity and a red shift from 430 to 600 nm were observed, suggesting the formation of larger aggregates corresponding to the peptide coupling product (Response Fig. 3). Similar aggregation behavior was confirmed using Confocal Microscopy in the amino acid mixture I with 2b, demonstrating that selective peptide bond formation with arginine promotes aggregation even under competitive conditions.

We have revised the manuscript as follows: "Peptide conversions were monitored via UPLC and assessed after the completion of the reaction, which occurred over 48 hours. To investigate the aggregation dynamics following selective peptide bond formation with R, the system was further characterized. Confocal microscopy of a mixture containing 10 mM 2b and 10 mM R in 0.6 M borate buffer (pH 9.1) revealed a transition from smaller aggregates formed by 2b alone to larger assemblies, indicating that the peptide coupling product between R and **2b** promotes aggregate growth (Supplementary Fig. 25). This observation was supported by fluorescence spectroscopy, which showed

a decrease in intensity and a red shift from 430 to 600 nm, suggesting changes in the arrangement of the aromatics (Supplementary Fig. 26). Similar aggregates were visualized using confocal microscopy in amino acid mixture I with 2b, further confirming that selective peptide bond formation with arginine promotes aggregation under competitive conditions (Supplementary Fig. 27).

The following figures have been added to the Supplementary Information of the manuscript.

Supplementary Figure 25: Time-dependent confocal microscopy images of reaction between 10 mM **2b** and 10 mM R, in 0.6 M borate buffer, pH 9.1.

Supplementary Figure 26: Time-dependent Fluorescence emission spectra of 10 mM **2b** and 10 mM R, in 0.6 M borate buffer, pH 9.1.

Supplementary Figure 27: Confocal microscopy images of reaction between 10 mM **2b** and 50 mM amino acid mixture 1 (D, S, L, R, F, each amino acid is at 10 mM concentration) measured after 48 hours, in 0.6 M borate buffer, pH 9.1.

Bottom of p. 5 and beginning of p. 6. The system discussion 'navigates' freely between mixtures containing R or K. There is no explanation for that... and it is not always clear when one ends and the other starts.

Response: We acknowledge that the transition between discussing mixtures containing arginine (R) and lysine (K) was not clearly demarcated. To address this, we have revised the text to explicitly state when we are referring to arginine or lysine in the discussion. We have also ensured that each section is clearly focused on one amino acid before transitioning to the other, providing clarity in the flow of the discussion. Additionally, in light of the reviewer's comments, we have conducted further experiments using ornithine (O) and diaminopropionic acid (Dpr) as models for lysine. These experiments confirmed the observed trends in peptide coupling, providing further support for the role of side chain reactivity in the coupling process.

We have revised the manuscript as follows: "Building on the selective coupling with R, we explored the behavior of lysine (K), another positively charged amino acid with distinct reactivity attributed to its flexible aliphatic side chain terminating in an ϵ -amine group. Replacing R with K in mixture I resulted in peptide bond formation with K, demonstrating the important role of electrostatic interactions in promoting selective coupling (Supplementary Figs. 29A, 30 and 31A). We noticed that lysine undergoes a two-step coupling process: first, an amide bond forms through the $N\alpha$ primary amine, followed by a second coupling through the reactive ϵ -amine. As expected, the use of trimethyl lysine eliminated side reactions from the ϵ -amine, while still maintaining selective peptide bond formation (Supplementary Figs. 29 B, 31B and 32-34). In competition experiments between R and K using 2b, we were unable to separate the coupling products using UPLC. Fluorescence microscopy revealed the absence of the typical spherical aggregates observed with high yields of arginine coupling products, suggesting that lysine was also incorporated (Supplementary Fig. 35). To overcome the analytical challenges with R and K, we introduced ornithine (O) and diaminopropionic acid (Dpr). Competition experiments between R and O, as well as R and Dpr, were carried out with both 2a and 2b. The results were consistent with those obtained for K, confirming side chain reactivity (Supplementary Fig. 36-45). Additionally, O and Dpr were tested by replacing K in amino acid mixture I, and similar trends in peptide coupling were observed, further supporting the role of side chain interactions in determining the outcome of these reactions (Supplementary Fig. 46-55)."

p. 6 line 187 and on. "...perform experiments with all 20 natural amino acids in one pot. Both for 2a and 2b, we observed that peptide coupling with cysteine (C) via thioester bond formation predominated over other amino acids (Fig. 2H)." This result is completely orthogonal to the paper's study. It is perfectly clear why Cys reacted faster... Since Cys reacts via NCL which is both efficient and selective, the analysis with Cys comprises a completely different aspect. The other aspects of the study with all 20 aas are described only in the SI. This is weird because they provide the most essential and interesting data.

Response: We appreciate the reviewer's insightful observation regarding cysteine's reactivity via native chemical ligation (NCL) and the distinct role of its highly nucleophilic thiol side chain. This unique behavior could indeed overshadow the coupling trends of the other amino acids, potentially causing confusion in the context of our study. Since the main focus of our work is the reactivity and self-assembly behavior of the phosphate esters, rather than the nucleophilicity of the amino acid side chains (e.g. thiols), we have decided, also in alignment with suggestions from other reviewers to exclude the experiments involving cysteine from this manuscript. However, this interesting cysteine behavior will be explored in more detail in a future study.

Incoherent/unjustified text and terminology.

P. 4 line 100. "integrate structural elements in their structure for preorganization into compartments". Instead of 'amphiphilic'.

Response: We have replaced the original phrase with the following "integrate structural elements in their structure that promote self-assembly."

- P. 4 line 104-5. "... where the phosphate esters act as solubility tags..". Instead of charged/polar residues increasing solubility.

Response: We have changed the term solubility tag, by using the phrase "enhance the solubility of hydrophobic amino acids residues in an aqueous environment.⁴⁷"

- P. 5 line 127. At the end of the introduction the manuscript mentions: "We also attempted experiments with all twenty natural amino acids in one pot, with the aim of determining the effect of amino acids with nucleophilic side chains on peptide bond formation." But, what did you get here? This is also not so clear when reading the results sections (see comments above).

Response: As mentioned earlier, since the experiments involving cysteine have been excluded from the manuscript, this text has been removed to avoid any confusion.

Reviewer:2

This work reported an amide bond formation strategy using aminoacyl phosphate esters. By tuning the phosphate ester structure, the peptide bond could be selectively formed according to the molecule electrostatic charge or aromaticity. Besides, processes including self-sorting and co-assembly could be controlled. This strategy is promising in sequential peptide coupling with selectivity and orthogonality. However, there are several issues requiring further elaboration before this work can be recommended for publication on Nature Communications.

Response: We thank the reviewer for recognizing the importance of our strategy for building selective peptide coupling and oligomerization, and for considering our manuscript for publication in Nature Communications following the required revisions.

1) The phrases “preorganization” and “preorganization into compartment” are quite confusing, since there is no extra process described before reaction with amino acids in the solution. Do they mean structural modification or structural regulation? Further elaboration on this issue is necessary. Exchange to easier-understanding words would be better.

Response: We appreciate the reviewer’s comment regarding the use of the term “preorganization.” We agree that the term may have caused confusion, as there is no preceding process taking place to justify its use. In this context, “preorganization” was intended to describe the inherent ability of the aminoacyl phosphate esters to adopt specific structural arrangements that facilitate selective peptide coupling and assembly prior to their consumption. To enhance clarity, we have revised the manuscript to replace “preorganization” with “self-assembly” and “aggregation” where appropriate. This more accurately reflects the process of phosphate esters forming specific structures in solution before reacting with amino acids.

2) The selectivity, or the departure ability from randomness, of the phosphate ester is suggested to be quantitatively determined by the thermodynamic reaction free energy, which is not hard to achieve. This would be more explicit and persuasive than the simple ratio percentages (Lines 141, 146, and 147).

Response: We thank the reviewer for the suggestion regarding the use of thermodynamic reaction free energy to quantify the selectivity of the phosphate ester system. In response, we calculated the equilibrium constant and Gibbs free energy. These calculations provide insights into the spontaneity of the reaction and support our findings on the favorable nature of peptide bond formation. The high conversion yields observed in our experiments, indicate the effectiveness of the coupling process. However, we believe that a more comprehensive determination of the thermodynamic parameters would require addressing complex mechanistic pathways, such as competing hydrolysis and side reactions, which are not captured by simple equilibrium calculations. Additionally, determining thermodynamic parameters in complex mixtures (amino acid mixture I or II), composed of many different components, presents significant challenges due to the multiple reaction pathways and interactions between species.^{3,4} These complexities have been discussed in previous literature, including the works of Tamura and Schimmel (2003) and Carpenter (1960), who explored similar challenges in phosphate ester systems and hydrolytic reactions.

We have calculated the thermodynamic free energy by using following equation:

$$\Delta G = -RT \ln K$$

For example, in case of 10 mM **2a** with 10 mM S, calculations give rise to $\Delta G = -0.45$ kJ/mol.

Given the high yields observed, the calculated Gibbs free energy values for the range of 7 to 9 mM coupling rank between approximately 0.62 kJ/mol and -5.44 kJ/mol, indicating increasing spontaneity with higher conversion.

3) Most of the aminoacyl phosphate esters designed in this work are non-natural amino acid derivatives. It is interesting how is the feasibility to use natural amino acids as the phosphate ester. Additional discussion on this issue is recommended.

We appreciate the reviewer's insightful comment regarding the use of natural amino acids as phosphate esters. In response, we synthesized three new derivatives incorporating alanine and valine (**2c**, **2d** and **1d**) to investigate whether aliphatic amino acids could impact the coupling process. Interestingly, we observed unexpected changes in reactivity when comparing aromatic and aliphatic amino acids, revealing an additional mechanism distinct from both the aromatic natural and non-natural derivatives used in this study.

Specifically, experiments with **2c** (Boc-AEP) demonstrated random selection, similar to **2a** phenylalanine, where the reactivity did not differ significantly. However, replacing alanine with a bulkier aliphatic side chain, such as valine, led to a significant decrease in reactivity and increased coupling with arginine. These results highlight the crucial role of the chemical nature of amino acid side chains in dictating the reactivity patterns of the phosphate esters. Additionally, the Fmoc derivative, which exhibited aggregation, showed the lowest reactivity and the strongest preference for reacting with arginine. This reduced reactivity of the Fmoc derivative, driven by its modified aromatic structure and its ability to form aggregates, was unexpected when compared to the parent **2a** molecule, which, despite maintaining its aromatic character, lacked similar self-assembly behavior.

The reviewer's feedback prompted us to consider an important question for our manuscript: how can the reactivity be slowed down to allow for more selective coupling and better control over the system's outcome? The new experiments identified two primary mechanisms that can be used to control reactivity, leading to enhanced selection:

- Interaction between amino acid side chains (*aliphatic vs. aromatic*): We discovered that the nature of the amino acid side chain plays a critical role in modulating reactivity. Aliphatic amino acids, such as alanine and valine, exhibited different reactivity patterns compared to their aromatic counterparts, such as phenylalanine, where an inductive effect from the aromatic rings increased reactivity. This difference in molecular structure allows for tunable control over reactivity, enabling selective coupling depending on the side chain's characteristics.
- *Self-assembly propensity of the amino acyl phosphate esters*: Another key mechanism we identified involves the self-assembly of the activated amino acids, particularly in the Fmoc-derivative, which showed a strong tendency to form aggregates. The ability of certain amino acids to self-assemble creates a more organized environment, slowing down the reactivity and providing a controlled setting for peptide bond formation. This behavior contrasts with **2a**, where

the lack of self-assembly led to faster and more random coupling events, despite the use of an aromatic amino acid.

Together, these two mechanisms offer a strategic approach to slowing down reactivity and enhancing selectivity. The interaction between amino acid side chains and the self-assembly potential of the system both act as levers to control the peptide coupling process. These insights represent an important step forward in achieving a more selective and tunable peptide bond formation process, as evidenced by our experimental results.

We have added the following test in the manuscript: “Having demonstrated the role of aromatic amino acids in the structure of phosphate esters in promoting aggregation and peptide coupling, we were prompted to explore the impact of other side chains on peptide bond formation within amino acid mixture I. Thus, we synthesized two additional derivatives, Boc-AEP (**2c**) and Boc-VEP (**2d**), to investigate how aliphatic residues could influence the process. Boc-AEP (**2c**) exhibited a moderate half-life of 4 hours similar to Boc-FEP (**2a**), whereas Boc-VEP (**2d**) showed a significantly longer half-life of 49.5 hours, highlighting the stabilizing effect of the bulkier aliphatic side chain (Fig. 2E, Supplementary Figs. 1c and 1d). These experiments revealed distinct differences in reactivity between aromatic and aliphatic amino acids. Aromatic side chains in aminoacyl phosphate esters exert inductive effects, increasing the electrophilicity of the carbonyl group, making it more prone to hydrolysis. However, in the presence of aggregates, this susceptibility is reduced as the activated amino acids become shielded from external nucleophiles, including water. **2c** showed a product distribution similar to **2a** with amino acid mixture I, as both displayed comparable reactivity (Supplementary Figs. 72-74). Interestingly, when alanine was replaced with valine (**2d**), reactivity significantly decreased, yet this led to enhanced peptide coupling with R (Supplementary Figs. 75-77). Time-dependent analysis of the libraries (Supplementary Fig. 78) revealed that **2b** exhibited slower kinetics compared to all other derivatives but demonstrated the highest selectivity. Although **2d** features an aliphatic side chain, it also facilitated selective peptide bond formation with R, albeit at a lower level than **2b**. Overall, these findings emphasize that both side chain interactions and self-assembly can work together to slow down reactivity while enhancing selectivity in peptide coupling.”

Supplementary Figure 1: Hydrolysis of 10 mM A) **2a**, B) **2b**, C) **2c** and D) **2d** in 0.6 M borate buffer, pH 9.1. The half-life of **2a**, **2b**, **2c** and **2d** is 2.17, 99, 4 and 49.5 hours respectively. The half-lives are calculated by using the equation: $t_{1/2} = 0.693/k$, assuming first-order kinetics. Error bars represent the standard deviation of three independent experiments. Inset photos represent the macroscopic behavior of the samples throughout the reactions.

Supplementary Figure 72 and 75: Peptide bond formation between A) **2c**, B) **2d** with amino acids mixture I (D, S, L, R, and F). The concentration used for **2c** or **2d** was 10 mM, while the total concentration for the amino acids in the mixture was 50 mM (10 mM each).

4) According to Figure 3C, the selective production of -R and -F(4-gua) is not remarkable, so that the schematic Figure 3B is somehow misleading. The Figure 3E and Figures 3FG are also contradict, because 3E describes a three-component mixture while 3FG compare only 2 fractions. Revision on these contents is requested. By the way, since the pairs of EP/PP and 2b/3 have the same structural difference, could EP/PP be exchanged to 2b/3 for clarity?

Response: We thank the reviewer for his/her suggestion and we have removed Figure 3b. Regarding Figures 3E and 3F, we understand the concern about the discrepancy between these figures. Figure 3E is indeed a schematic of a three-component mixture, while Figure 3F focuses only on the two dominant components due to negligible coupling with DD, which is presented in Supplementary Fig. 104. To avoid further confusion, we have revised Figure 3 to display only two-component mixture experiments. Additionally, we have replaced the labels for **2b** and **3** with EP/PP, as suggested, to enhance clarity and consistency throughout the figure.

Fig. 3: Effect of phosphate ester on peptide coupling. **A**) Schematic representation of the interactions involved between aminoacyl phosphate ester (**3**) and a phenylalanine amino acid residue with a substitution of a guanidine group in the para position. Peptide bond formation between **2b** or **3** with **B**) 10 mM F(4-guanidine) and **C**) 10 mM F. The concentration used for **2b** or **3** was 10 mM and the samples were prepared in 0.6 M borate buffer pH 9.1. Peptide coupling was measured after 48 hours. **D**) Schematic representation of a competition experiment between **2b** or **3** and a dipeptide mixture containing DR and DW. Peptide bond formation between **2b** or **3** with mixture of dipeptides containing **E**) DR and DW (10 mM each) and **F**) DD and DW (10 mM each) in 0.6 M borate buffer pH 9.1. The concentration used for **2b** or **3** was 10 mM. In each bar graph, the **2b** and **3** represent ethyl and phenyl phosphate, respectively. The x axes indicate the sequence of peptide bonds formed when compounds **2b** or **3** are used with individual amino acids and dipeptides. The bars with striped lines represent the hydrolysis product (**3-OH** or **2b-OH**). Error bars represent the standard deviation of three independent experiments. Peptide coupling yields were measured at the end reaction i.e. after 72 hours. **G**) Cryo-Electron Microscopy images (from left to right) of 10 mM **3** immediately after dissolving, after 80 minutes, after complete hydrolysis (15 days) and upon coupling with 10 mM DW.

5) Covalent self-sorting of **1b** and **1e** without any production of mixed sequence is amazing. A deeper analysis on the mechanism in addition to the hydrophobicity/hydrophilicity is highly encouraged.

Response: We thank the reviewer for appreciating the covalent self-sorting results. In the original submission, we demonstrated covalent self-sorting by mixing **1g** and **1b** (Figure 4C). Following the reviewer's previous comment on the effect of natural amino acid derivatives, we took the opportunity to further explore the mechanistic basis of covalent self-sorting. To expand on the generality of this approach, we have now included one additional example of VEP (**1d**) (Supplementary Figs. 136-139, Supplementary Table 6) in the revised manuscript. Our findings suggest that the self-sorting mechanism is closely linked to the aggregation behavior of the aminoacyl phosphate esters. Derivatives such as **1a**, **1d**, and **1g**, which do not aggregate, resulting in oligomerization occurring predominantly in the solution phase. In contrast, **1b** undergoes oligomerization in the aggregated phase, leading to a non-mixing environment and covalent self-sorting. This difference in phase behavior plays a crucial role in driving the covalent self-sorting process. Confocal microscopy further confirmed that **1b** formed spherical aggregates that transitioned into fibrillar structures over time, whereas **1g** remained in solution without forming aggregates (Supplementary Fig. 130). Fluorescence spectroscopy also supported these findings, as the emission spectra of mixed solutions containing **1b** and **1g** showed no change compared to **1b** alone, indicating that phase separation prevented interaction between the species (Supplementary Fig. 131).

To further support this mechanism, we performed additional centrifugation experiments, which revealed the formation of two distinct phases, each selectively enriched in specific amino acid compositions. This observation reinforces the idea that phase behavior plays a critical role in covalent self-sorting. The phase containing aggregated **1b** was predominantly enriched in oligomers derived from this derivative, while the solution phase, where **1g** remained unaggregated, was enriched in its respective oligomeric products. This selective distribution between the phases provides strong evidence for the role of aggregation in promoting self-sorting and preventing the formation of mixed sequences.

We have updated the self-sorting section in the manuscript as follows: "Covalent self-sorting versus co-assembly: Building on our previous observations with N-terminus protected aminoacyl phosphate esters, we next explored peptide oligomerization using N-terminus free aminoacyl phosphate esters. We hypothesized that assembly of aminoacyl phosphate esters might allow for controlled oligomerization (Fig. 4A). Thus, we synthesized derivatives incorporating aromatic amino acid residues: FEP (**1a**), F(4-NH-Fmoc)EP (**1b**), F(4-guanidine)EP (**1e**), BPAEP (**1f**) and F(4-COOH)EP (**1g**). These activated amino acids were synthesized as ethyl esters. When **1b** and **1f** were mixed at equimolar concentrations (10 mM each) in 0.6 M borate buffer (pH 9.1), we observed the formation of mixed oligomeric species (Fig. 4B, Supplementary Figs. 115-117 and Supplementary Table 2). Microscopy studies revealed that both **1b** and **1f** formed spherical aggregates, similar to what was observed with previously studied protected aminoacyl phosphate esters like **2b** and **3** (Supplementary Figs. 118,119). Although no hydrogel formation was detected in the BPA-containing libraries, a weak hydrogel developed in the Fmoc-containing samples after oligomerization. Rheological studies further suggested that the incorporation of mixed sequences enhanced the mechanical properties of the resulting structures, pointing towards a co-assembly effect between oligomers (Supplementary Figs. 120, 121). The role of guanidine residues was also significant in these systems. When guanidine was present (**1e**), it promoted the formation of hetero-oligomers through electrostatic interactions with the phosphate esters, highlighting its importance in driving co-assembly and influencing the organization of mixed species (Supplementary Figs. 122-125 and Supplementary Table 3).

Next, we explored the behavior of libraries containing **1b** and **1g**. This mixture resulted in two distinct families of oligomers enriched in either Fmoc- or F(4-COOH)-containing species, indicating covalent self-sorting (Fig. 4C, Supplementary Figs. 126-128 and Supplementary Table 4). Centrifugation

experiments revealed distinct phases, where the aggregated phase was predominantly enriched with **1b** oligomers, while **1g** oligomers remained in the solution (Supplementary Fig. 129). Confocal microscopy further confirmed that **1b** formed spherical aggregates that transitioned into fibrillar structures over time, whereas **1g** remained in solution without forming aggregates (Supplementary Fig. 130). Fluorescence spectroscopy also supported these findings, as the emission spectra of mixed solutions containing showed no change compared to **1b** alone, indicating that phase separation prevented interaction between the species (Supplementary Fig. 131). Self-sorting was also observed in libraries containing **1b** and **1a** (Supplementary Figs. 132-135 and Supplementary Table 5). To confirm the generality of this process, we extended the system to aliphatic aminoacyl phosphate esters, VEP (**1d**) (Supplementary Figs. 136-139 and Supplementary Table 6). Overall, these findings suggest that the assembly of aminoacyl phosphate esters creates a protective microenvironment that positions primary amines in close proximity to the acyl phosphate esters, facilitating selective coupling and homo-oligomerization. However, when additional non-covalent interactions, such as aromatic stacking (as seen with **1f**) or electrostatic forces (as observed with **1e**), are introduced, the aggregates engage in co-assembly, leading to the formation of hetero-oligomers. This balance between covalent self-sorting and co-assembly offers a versatile strategy for controlling peptide oligomerization, allowing specific amino acid residues to interact preferentially within different phases (aggregated vs. solution), ultimately driving the selective formation of oligomeric species with distinct chemical compositions.”

Additional figures in the Supplementary Information of the manuscript have been added.

Fig. 4: Covalent self-sorting and co-assembly in spontaneous oligomerization reactions. A) Schematic representation of the oligomerization reaction between **1b** and other amino acyl phosphate esters (**1a**, **1d**, **1e**, **1f** and **1g**) resulting in covalent self-sorting or co-assembling. Formation of homo and hetero-oligomers between 10 mM **1b** with **B)** 10 mM **1f** and **C)** 10 mM **1g**. Libraries were prepared in 0.6 M borate buffer, pH 9.1. Peptide coupling was measured after 24 hours. Bar graphs represent the sum of all library components (grouped as homo or hetero-oligomers) formed in the reactions.

Supplementary Figure 137: UPLC chromatograms of A) 10 mM **1b** B), 10 mM **1d** and C) 10 mM **1b** with 10 mM **1d** (1:1), in 0.6 M borate buffer, pH 9.1.

Supplementary Figure 130: Time-dependent confocal microscopy images of A) 10 mM **1b**. B) 10 mM **1g** in 0.6 M borate buffer, pH 9.1.

Supplementary Figure 131: Time-dependent Fluorescence emission spectra of A) 10 mM **1b** and B) Mixed library of 10 mM **1b** and 10 mM **1g** in 0.6 M borate buffer, pH 9.1.

Supplementary Figure 129: UPLC chromatograms of mixed library of 10 mM **1b** and 10 mM **1g** after centrifugation A) Oligomers of **1b** in aggregated phase B) Oligomers of **1g** in solution phase.

6) The English written requires further polishing.

Response: We have thoroughly reviewed and polished the manuscript to improve clarity and overall readability.

Minor issues:

1) In Figure 1A, the R1 and R2 group of compound 3 has no definition.

Response: To avoid confusion, we have removed the numbers under the ethyl and the phenyl chemical structures.

2) In Figure 2A, the R group of compound 2a/2b-X should be X.

Response: We thank reviewer for pointing this out. We have now corrected this in Figure 2A.

Reviewer: 3

Comments: In this manuscript, Sharma et al. discuss how chemical modifications designed on aminoacyl phosphate esters affect peptide synthesis. The authors observe that specific activated substrates result in supramolecular structures that affect product distribution. The manuscript contains exciting results, including the control over the formation of hetero or homo-oligomers upon the design of the reactants. However, I found the manuscript not well-balanced: a too-long introduction that doesn't go deep enough on what's the state of the art in selective peptide synthesis in water, a very superficial characterization of the supramolecular assemblies, and no effort to explain the effect of different supramolecular assemblies on the results; a discussion that claims selective peptide formation when the reactions reported are not as selective as the authors claim; a final result (long oligomers synthesis) that would have deserved more than half a page and a more thorough explanation and characterization.

Response: We thank the reviewer for his/her thoughtful comments and for recognizing the potential of our work. We have carefully considered the points raised and have made significant revisions to improve our manuscript.

- *Introduction:* We agree with the reviewer that the introduction would benefit from a discussion on selective peptide coupling. In revising the introduction, we have included examples where self-assembly impacted the properties of peptide bond formation. While there are numerous methods for peptide synthesis, such as the use of carboxy anhydrides and other strategies that operate in organic solvents, our aim was not to provide a comprehensive review of all peptide bond formation techniques. Instead, we focused on those examples that align with our research, particularly highlighting cases where self-assembly played a significant role in controlling reactivity and selectivity in aqueous environments. This revision ensures that our study is positioned within relevant advancements, while emphasizing the unique aspects of abiotic phosphate-driven peptide coupling and self-sorting.

We added the following text in the manuscript: “In the context of amide bond formation,^{34, 35, 36} selective peptide coupling has been demonstrated using redox-active coacervates,³⁷ which facilitate controlled reactivity by creating microenvironments that stabilize intermediates and enhance specificity. Similarly, surfactant aggregates³⁸ have been shown to drive spontaneous polypeptide formation from aminoacyl adenylates,^{39, 40} highlighting the role of self-assembled systems in oligomerization reactions. In other strategies, wet-dry cycles⁴¹ and metal ions⁴² have been employed to induce peptide elongation in aqueous media, though these conditions are often incompatible with the formation of stable self-assembled structures. Additionally, enzymatic methods⁴³, thioester formation⁴⁴ and the use of N-carboxy anhydrides (NCAs)⁴⁵ have allowed for the coupling of specific amino acid residues, though these strategies face challenges in solubility, reactivity control and selectivity over self-assembling motifs.”

- *Characterization of Supramolecular Assemblies:* We acknowledge that the original manuscript did not provide a thorough characterization of the supramolecular assemblies. In response, we have expanded this section to include additional experiments that more clearly characterize the assemblies formed by the aminoacyl phosphate esters. We have also incorporated microscopy (fluorescence and electron microscopy) and spectroscopic techniques (fluorescence, dynamic light scattering) to provide a more detailed understanding of the nature of these assemblies and their role in influencing product distribution.
- *Effect of Supramolecular Assemblies on Product Distribution:* We have clarified the reaction processes that occur both with and without amino acid mixtures, along with the formation of

supramolecular assemblies at different stages of the reaction. Specifically, we have observed the formation of spherical aggregates from the aminoacyl phosphate esters, followed by fibrillar structures upon hydrolysis of the product. We have added a more detailed analysis of how these distinct supramolecular assemblies influence the reaction outcomes, particularly in relation to product selectivity and distribution.

- *Long Oligomer Synthesis:* We agree that the synthesis of long oligomers required more detailed explanation and characterization. To address this, we have expanded this section, providing additional data on the specific conditions under which longer oligomers are formed. Furthermore, we have included a more thorough analysis of the self-sorting mechanism that drives oligomerization. Additional experiments, such as centrifugation, have been performed to support our conclusions. We have also added more examples demonstrating the generality of the approach, alongside fluorescence experiments that further elucidate the role of supramolecular assemblies in guiding the formation of long oligomers.

1) Figure 1 is confusing, as it comes too early in the manuscript when the readers haven't yet read about all the results. There are too many catchy words, with not enough context because everything will be explained much later in the text. I suggest moving this figure to the end, as it helps summarize the results.

Response: We have modified Figure 1 as follows to focus on introducing the design of the aminoacyl phosphate esters and their role in driving selective peptide bond formation. The other parts of the figure, particularly the schematics, have been moved to the relevant sections that describe the experimental findings in greater detail.

Fig. 1: Aminoacyl phosphate ester-mediated peptide coupling in aqueous media: A) Chemical reaction of peptide bond formation between aminoacyl phosphate esters and amino acids in borate buffer (0.6 M, pH 9.1). **B)** Chemical structures of aminoacyl phosphate esters, highlighting the different parts of the molecule (N-terminus, amino acid side chains and phosphate esters) that can be tailored

to investigate spontaneous peptide bond formation. Structures of the aminoacyl phosphate esters used in the study feature variations in side chains ($R^2 = a-g$) and phosphate ester groups ($R^3 = \text{ethyl and phenyl}$). These modifications allow systematic investigation into how the side chains and phosphate ester groups influence peptide formation and reaction kinetics.

2) In the experiments with all 20 amino acids, the predominant product is always the Cy-containing peptide. Even though the authors show they can selectively control the formation of the other products, they do not have any control over the main product. This result seems to undermine the whole point of the manuscript while not adding much. I would consider removing the set of experiments.

Response: We appreciate the reviewer's comment regarding the predominance of cysteine-based coupling in the experiments with all 20 amino acids. We agree that cysteine's reactivity, driven by its highly nucleophilic thiol group, dominates these reactions and can overshadow the formation of other products. While this behavior is interesting, it shifts the focus away from the primary goal of our manuscript, which is to explore the reactivity and self-assembly behavior of the phosphate esters, rather than the nucleophilicity of the amino acid side chains (e.g., thiols). In alignment with the reviewer's suggestion, we have decided to remove this set of experiments to maintain the focus on how our system achieves selective peptide coupling through phosphate ester design and supramolecular assembly. This adjustment ensures that the manuscript remains centered on its key findings.

3) Tables in the SI (for example, Tables 2 and 3) would be more informative if yields/product distribution were included. Also, yields are rarely given in the main text, leaving the readers to guess what worked better by comparing the height of bars in the graphs.

Response: Following the reviewer's previous suggestion and since we have removed the experiments involving 2a and 2b with all 20 amino acids, the associated Tables 2 and 3 have also been removed, as they specifically referenced these experiments.

4) The authors report the selection for peptide bonds formed with R and K within mixtures. R and K are rather different positively charged amino acids, including the ability of R, but not of K, to interact with aromatics. The readers are thus left wondering if there is any preference or competition between R and K when they are both present as reactants. The manuscript would gain in depth if the authors would perform this experiment (and explain the results).

Response: We thank the reviewer for the insightful comment. Both lysine (K) and arginine (R) are positively charged amino acids that exhibit distinct chemical properties that influence their behavior in peptide coupling and interactions. Lysine, with its flexible aliphatic side chain terminating in an ϵ -amine group, can participate in both first-step (α -amine) and second-step (ϵ -amine) coupling reactions. Arginine, on the other hand, has a guanidinium group that has been shown for its ability to form strong cation- π interactions with aromatic groups and electrostatic interactions with phosphates. In our experiments, we found that lysine's second-step coupling through the ϵ -amine consistently outcompetes its first-step coupling via the α -amine in both mixture and individual reactions. This is due to the increased nucleophilicity of the ϵ -amine, which leads to higher reactivity compared to the guanidinium group of arginine. Despite significant efforts, we were unable to separate the coupling products of arginine and lysine in a mixture using UPLC. However, competition experiments between R and K with 2b using fluorescence microscopy revealed the absence of the spherical aggregates typically observed when arginine-containing coupling products are formed in high yield. This absence suggests that the predominant coupling products in these experiments involve lysine, as indicated by the altered aggregation patterns.

Supplementary Figure 35: Confocal microscopy images of reaction between A) 10 mM **2b** and 10 mM R, B) 10 mM **2b** and 20 mM amino acid mixture of R and K (10 mM each) in 0.6 M borate buffer, pH 9.1.

To gain further insights and achieve better separation between lysine and arginine derivatives, we have employed ornithine and diaminopropionic acid as alternative models. These results will be discussed in the following comment.

5) Additionally, the authors cite Powner's work on the selective synthesis of Lys-containing peptides. Have the authors tested ornithine and diaminopropionic acid in place of lysine? What results would the authors expect?

Response: We have performed experiments using ornithine and diaminopropionic acid in place of lysine. Specifically, we conducted competition experiments between arginine and ornithine, as well as arginine and diaminopropionic acid, with both **2b** and **2a**. Our findings with ornithine and diaminopropionic were consistent with those for lysine, involving side chain coupling.

Overall, from this and the previous comment from the same reviewer, we have revised the manuscript as follows: "Building on the selective coupling with R, we explored the behavior of lysine (K), another positively charged amino acid with distinct reactivity attributed to its flexible aliphatic side chain terminating in an ϵ -amine group. Replacing R with K in mixture I resulted in peptide bond formation with K, demonstrating the important role of electrostatic interactions in promoting selective coupling (Supplementary Figs. 29A, 30 and 31A). We noticed that lysine undergoes a two-step coupling process: first, an amide bond forms through the $N\alpha$ primary amine, followed by a second coupling through the reactive ϵ -amine. As expected, the use of trimethyl lysine eliminated side reactions from the ϵ -amine, while still maintaining selective peptide bond formation (Supplementary Figs. 29 B, 31B and 32-34). In competition experiments between R and K using **2b**, we were unable to separate the coupling products using UPLC. Fluorescence microscopy revealed the absence of the typical spherical aggregates observed with high yields of arginine coupling products, suggesting that lysine was also incorporated (Supplementary Fig. 35). To overcome the analytical challenges with R and K, we introduced ornithine (O) and diaminopropionic acid (Dpr). Competition experiments between R and O, as well as R and Dpr, were carried out with both **2a** and **2b**. The results were consistent with those obtained for K, confirming side chain reactivity (Supplementary Fig. 36-45). Additionally, O and Dpr were tested by replacing K in amino acid mixture I, and similar trends in peptide coupling were observed, further supporting the role of side chain interactions in determining the outcome of these reactions (Supplementary Fig. 46-55)."

In the Supplementary Information the following figures have been added.

Supplementary Figure 40: Bar graphs showing peptide coupling between A) **2a**, B) **2b** with 20 mM amino acid mixture of R and O (10 mM each) in 0.6 M borate buffer, pH 9.1. In each bar graph striped bar represent the hydrolysis product **2a-OH/2b-OH**. Error bars represent standard deviation from three independent experiments. Peptide coupling yields were measured for **2a**, after 30 minutes, while for **2b**, after 48 hours.

Supplementary Figure 45: Bar graphs showing peptide coupling from competition experiments between A) **2a**, B) **2b** with an amino acid mixture containing 10 mM R and 10 mM Dpr in 0.6 M borate buffer, pH 9.1. In each bar graph striped bar represent the hydrolysis product **2a-OH/2b-OH**. Error bars represent standard deviation from three independent experiments. Peptide coupling yields were measured for **2a**, after 30 minutes, while for **2b**, after 48 hours.

Supplementary Figure 50: Bar graphs showing peptide coupling between A) 10 mM **2a** or B) 10 mM **2b** with 50 mM amino acid mixture (D, S, L, O, F, each amino acid is 10 mM), in 0.6 M borate buffer, pH 9.1. In each bar graph striped bar represent the hydrolysis product **2a-OH/2b-OH**. Error bars represent standard deviation from three independent experiments. Peptide coupling yields were measured for **2a** after 30 minutes, while for **2b** after 48 hours.

Supplementary Figure 55: Bar graphs showing peptide coupling between A) 10 mM **2a** and B) 10 mM **2b** with 50 mM amino acid mixture (D, S, L, Dpr, and F, each amino acid is 10 mM), in 0.6 M borate buffer, pH 9.1. In each bar graph striped bar represent the hydrolysis product **2a-OH/2b-OH**. Error bars represent standard deviation from three independent experiments. Peptide coupling yields were measured for **2a** after 30 minutes, while for **2b** after 48 hours.

6) Figure 2, panel E: why is F more efficient at forming peptide bonds in ACN compared to L? And why is F so efficient in the presence of sodium sulfate?

Response In ACN, phenylalanine (F) may exhibit better solubility compared to leucine (L) due to its aromatic side chain, which interacts more favorably in organic solvents. While hydrophobic interactions are typically reduced in ACN, phenylalanine's enhanced solubility allows it to remain more available for reaction, thus facilitating more efficient peptide bond formation. On the other hand, leucine, with its aliphatic side chain, shows poorer solubility in ACN, which likely contributes to lower peptide coupling efficiency. Regarding phenylalanine's improved coupling efficiency in the presence of sodium sulfate, this can be attributed to the "salting-out" effect, wherein the solubility of hydrophobic amino acids like phenylalanine is reduced in aqueous environments. Sodium sulfate promotes aggregation or phase separation of hydrophobic molecules, concentrating the reactants locally and thereby increasing the likelihood of peptide bond formation. This localized phase behavior boosts phenylalanine's coupling efficiency by enhancing reactivity through the local concentration effect. To further clarify these findings, we performed an additional experiment investigating the competition between F and L (without the involvement of other amino acids) under two experimental conditions: organic solvent (ACN) and sodium sulfate. The results show a similar trend of F being more reactive in both conditions, though hydrolysis of the phosphate esters was observed in the presence of salts. However, we acknowledge that while these results might suggest that solubility and salting-out effects contribute to the observed selectivity, we cannot fully exclude other mechanisms, such as those involving Hofmeister series effects or other phenomena.

Figure: Bar graph showing Comparison of peptide bond formation between 10 mM **2b** and amino acid mixture containing F and L, (each amino acid is 10 mM), in 80 % ACN and 1M Sodium sulphate.

7) It would be helpful if the authors could state more clearly in the main text when the samples were analyzed by UPLC, considering that the aggregates' morphologies change over time. Does the product distribution change if the sample is analyzed after 10 minutes, 1 hour or 10 days?

Response: We have clarified in the main text the specific time points at which the samples were analyzed by UPLC, and this information has now been added to all relevant figures. Following the reviewer's suggestion, we have now monitored the reactions of **2a**, **2b**, **2c** and **2d** with the amino acid mixture I over time, analyzing the product distribution.

We have revised the manuscript as follows: "Having demonstrated the role of aromatic amino acids in the structure of phosphate esters in promoting aggregation and peptide coupling, we were prompted to explore the impact of other side chains on peptide bond formation within amino acid mixture I. To this end, we synthesized two additional derivatives, Boc-AEP (**2c**) and Boc-VEP (**2d**), to investigate how aliphatic residues could influence the process. Boc-AEP (**2c**) exhibited a moderate half-life of 4 hours

similar to Boc-FEP (**2a**), whereas Boc-VEP (**2d**) showed a significantly longer half-life of 49.5 hours, highlighting the stabilizing effect of the bulkier aliphatic side chain (Fig. 2E, Supplementary Figs. 1c and 1d). These experiments revealed distinct differences in reactivity between aromatic and aliphatic amino acids. Aromatic side chains in aminoacyl phosphate esters exert inductive effects, increasing the electrophilicity of the carbonyl group, making it more prone to hydrolysis. However, in the presence of aggregates, this susceptibility is reduced as the activated amino acids become shielded from external nucleophiles, including water. Boc-AEP (**2c**) showed a product distribution similar to Boc-FEP (**2a**) with amino acid mixture I, as both displayed comparable reactivity (Supplementary Figs. 72-74). Interestingly, when alanine was replaced with valine (**2d**), reactivity significantly decreased, yet this led to enhanced peptide coupling with R (Supplementary Figs. 75-77). Time-dependent analysis of the libraries (Supplementary Fig. 78) revealed that **2b** exhibited slower kinetics compared to all other derivatives but demonstrated the highest selectivity. Although **2d** features an aliphatic side chain, it also facilitated selective peptide bond formation with R, albeit at a lower level than **2b**. Overall, these findings emphasize that both side chain interactions and self-assembly can work together to slow down reactivity while enhancing selectivity in peptide coupling.”

The following figure has been added to the Supplementary Information.

Supplementary Figure 78: Time-dependent peptide bond formation between A) **2a**, B) **2b**, C) **2c**, D) **2d** with amino acids mixture I (D, S, L, R, F) in 0.6 M borate buffer, pH 9.1. The concentration used for **2a**, **2b**, **2c** and **2d** was 10 mM, while the total concentration for the amino acids in the mixture was 50 mM (10 mM each).

8) Many TEM and CryoEM images in the paper are confusing. At times, there are dark black spheres, at times, droplets, and at times, fibers. Are these samples amorphous or (partly) crystalline? The authors report DLS data that point towards tiny aggregates (10 nm), but the size of the aggregates observed by EM is very different.

Response: The structures observed change over time as the hydrolysis of the aminoacyl phosphate ester proceeds, which explains the different morphologies visualized, such as dark black spheres, droplets, and fibers.

This behavior has been observed both macroscopically and microscopically. The DLS data provide an average size of the aggregates, initially measured between 10 to 15 nm for **2b** when analyzed immediately (within 1 minute), which correlates well with the cryo-EM images captured at the early stages of the reaction. As the reaction progresses, the aggregate size increases, and the morphology evolves from small spherical aggregates to larger structures like fibers, which are visible in the later stages of the reaction. In light of the reviewer's concern, we have repeated the cryo-EM experiment for **2b** immediately after dissolving it in buffer. The results reproduced similar spherical aggregates (see below). To determine the aggregate size, we manually measured individual aggregates using image analysis software and fitted the data to normal and log-normal distribution curves (see below), resulting in an average size of 11.6 nm, with a range between 6 to 19 nm. These measurements are consistent with the DLS data taken at the early stages of the reaction. Similarly, for molecule **3**, we used this approach and found that its aggregates range in size from 7 to 17 nm, with an average size of 18 nm. This method provided consistent results with our earlier DLS studies, further supporting the dynamic nature of the aggregation process as the reaction proceeds.

Figure: Additional cryo-Electron Microscopy image of 10 mM **2b** immediately after dissolving in 0.6 M borate buffer, pH 9.1

Figure: Graphs showing A) Normal, B) Lognormal size distribution curve of aggregates of **2b** C) Statistical analysis of graph A and B showing average size and size range of aggregates of **2b**.

Figure: Graphs showing A) Normal, B) Lognormal size distribution curve of aggregates of **3** C) Statistical analysis of graph A and B showing average size and size range of aggregates of **3**.

9) Overall, the characterization of the aggregates is very superficial, yet the authors try to extrapolate information about the number of layers present in the aggregates. Light microscopy would have probably been a better choice for the characterization of the aggregates. The authors should also explain the inconsistencies in the size of the aggregates. Finally, the authors make no effort to hypothesize why aggregates (and which ones) control the selectivity of peptide bond formation. For example, is the aggregate of the aminoacyl phosphate ester the one that controls the selectivity, or is the co-assembly of the aminoacyl phosphate ester with the amino acid (like R) that controls it?

Response: We agree with the reviewer that our initial characterization was not sufficiently detailed. To address this, we have now incorporated fluorescence spectroscopy experiments to provide better insights into the arrangement of aromatic groups within the aggregates. This type of characterization is commonly employed in systems such as peptide amphiphiles, where aromatic residues can engage in π - π stacking interactions.⁵ Time-dependent fluorescence emission spectroscopy gave us insights into the aggregation behavior of **2b** and **3**. Initially, within 1 minute of mixing, we observed a broad emission peak at 323 nm. As the hydrolysis of **2b** proceeded over time, a notable decrease in fluorescence intensity and the appearance of a shoulder peak at 374 nm suggested a transition to other type of aggregates, which was further confirmed by time-dependent Confocal Microscopy. Additionally, we observed a significant decrease in fluorescence intensity (about 8 times) when comparing low concentrations of **2b** (1 mM, where it is mostly in a monomeric state) to higher concentrations (10 mM, where it is aggregated). This suggests that at low concentrations, the π - π stacking interactions between aromatic groups are weak, but as **2b** starts to aggregate at higher concentrations, these interactions become stronger and more pronounced. These findings align with the time-dependent size distribution measurements obtained from Dynamic Light Scattering (DLS) for **2b**, which also demonstrated a shift from smaller to larger aggregates over time.

Supplementary Figure 24: Fluorescence emission spectra of A) 10 mM **2b** at different time points. B) Zoomed in spectra of 10 mM **2b** at 60 min. C) Concentration-dependent fluorescence emission spectra of **2b** in 0.6 M borate buffer, pH 9.1.

Supplementary Figure 22: Time-dependent confocal microscopy images of 10 mM **2b**, in 0.6 M borate buffer, pH 9.1.

Supplementary Figure 22: Time-dependent size distribution measured by DLS of 10 mM **2b** in 0.6 M borate buffer, pH 9.1.

Similar trend of Fluorescence emission spectroscopy has been observed for **3**.

Supplementary Figure 114: Fluorescence emission spectra of A) 10 mM **3**. B) Zoomed in spectra of 10 mM **3** at 60 min. C) Concentration-dependent fluorescence emission spectra of **3**, in 0.6 M borate buffer, pH 9.1.

Based on our observations, we propose that the selectivity of peptide bond formation is governed by both the aggregation of the aminoacyl phosphate esters and their co-assembly with specific amino acids, such as arginine. The initial aggregation of the aminoacyl phosphate esters creates a structured microenvironment that can preorganize reactive species, enhancing selective interactions. Additionally, co-assembly between the aminoacyl phosphate esters and certain amino acids, facilitated by electrostatic interactions (in the case of positively charged residues like arginine) or π - π stacking (in the case of aromatic residues), likely further enhances selectivity.

We have added the following text in the manuscript: “To investigate the aggregation dynamics following selective peptide bond formation with R, the system was further characterized. Confocal microscopy of a mixture containing 10 mM 2b and 10 mM R in 0.6 M borate buffer (pH 9.1) revealed a transition from smaller aggregates formed by 2b alone to larger assemblies, indicating that the peptide coupling product between R and 2b promotes aggregate growth (Supplementary Fig. 25). This observation was supported by fluorescence spectroscopy, which showed a decrease in intensity and a red shift from 430 to 600 nm, suggesting changes in the arrangement of the aromatics (Supplementary Fig. 26). Similar aggregates were visualized using confocal microscopy in amino acid mixture I with 2b, further confirming that selective peptide bond formation with arginine promotes aggregation under competitive conditions (Supplementary Fig. 27). Ratio and pH-dependent experiments of 2b with amino acid mixture I revealed a direct competition between coupling with R and hydrolysis of the activated amino acid (Supplementary Figure 28).”

The following figures have been added to the Supplementary Information of the manuscript.

Supplementary Figure 25: Time-dependent confocal microscopy images of reaction between 10 mM 2b and 10 mM R, in 0.6 M borate buffer, pH 9.1.

Supplementary Figure 26: Time-dependent Fluorescence emission spectra of reaction between 10 mM 2b and 10 mM R, in 0.6 M borate buffer, pH 9.1.

Supplementary Figure 27: Confocal microscopy images of reaction between 10 mM **2b** and 50 mM amino acid mixture 1 (D, S, L, R, F, each amino acid is at 10 mM concentration) measured after 48 hours, in 0.6 M borate buffer, pH 9.1.

10) In the dipeptide experiments, why did the authors select tryptophan? It seems odd, considering they haven't tested *W* before. Why not use *DF* if the authors aim to test the effect of aromatics? Also, why did the authors change the amino acid at the C-term instead of at the N-term? Does the order have any influence?

Response: We thank the reviewer for the comment. The choice of tryptophan (*W*) was driven by its ability to engage in both ion- π and hydrophobic interactions, particularly through its indole ring, which has been shown to stabilize supramolecular assemblies and enhance coupling efficiency. This phenomenon is discussed in literature.⁶ Additionally, we have now conducted the experiment using *DF*, as suggested by the reviewer, and observed improved coupling efficiency with **3** compared to **2b**, supporting the hypothesis that aromatic residues enhance selective coupling, which is driven by interactions with the phosphate esters. By introducing aromatic or cationic residues in dipeptide sequences, we aimed to counteract the limitation and improve selectivity and coupling efficiency with sequences featuring aspartic acid residues. Our results, as shown in Figures 3F and 3G, demonstrate that this approach was effective. In response to the reviewer's suggestion, we also performed additional experiments comparing N-terminus modifications with *RD* and *DR* dipeptides. We observed that *RD* gave approximately 5% higher coupling efficiency than *DR*.

We added the following text in the manuscript: "Having established the effect of single amino acids on peptide bond formation, we next investigated how dipeptide sequences influence this process. The motivation for these experiments was to evaluate whether peptide substrates, could influence selectivity and overcome the limitations observed with negatively charged amino acids like aspartic acid. We started by examining the dyad of aspartic acid (*DD*), which, as expected, exhibited only trace amounts of peptide bond formation with both **2b** and **3** (Supplementary Figs. 96, 97). However, replacing the C-terminal aspartic acid with arginine (*DR*) and tryptophan (*DW*) significantly increased peptide coupling yields (Supplementary Figs. 98-101). We further explored mixtures containing two dipeptide sequences, including *DR*, *DW* and *DD*, *DW*. In these mixtures, **2b** (ethyl phosphate ester) primarily led to hydrolysis as the dominant product. In contrast, **3** (phenyl phosphate ester) favored the formation of *DW* as the predominant product (Figs. 3D, 3E and 3F). This trend remained consistent even in mixtures containing three dipeptide sequences (*DD*, *DR* and *DW*) (Supplementary Figs. 102-106). These results suggest that the aromatic nature of the phenyl phosphate head group in **3** enhances selectivity, particularly for aromatic residues like tryptophan. To further confirm this selectivity, we tested another mixture containing *DD* and *DF* with **2b** and **3**. The results revealed a similar behavior, where *DF* showed selective coupling with **3**, while **2b** predominantly resulted in hydrolysis (Supplementary Figs. 107-109). Notably, the high peptide bond conversion observed with *DW* suggests that the indole ring of tryptophan enhances stability through ion- π and hydrophobic interactions, improving the efficiency of peptide coupling and stabilizing the supramolecular assemblies.⁵⁶

The following figures have been added to the Supplementary Information of the manuscript.

Supplementary Figure 109: Bar graphs showing peptide coupling between 10 mM **2b** or 10 mM **3** with 20 mM dipeptides mixture of DD and DF (10 mM each) in 0.6 M borate buffer, pH 9.1. Peptide coupling yields were measured after 72 hours.

Figure: Bar graphs showing peptide coupling between A) 10 mM **2b** and 10 mM DR, B) 10 mM **3** and 10 mM RD, in 0.6 M borate buffer, pH 9.1.

11) Figure 3, panel A: I found it confusing that the authors highlight the electrostatic interaction between Arg and the phosphate (on the left side). This same interaction should also occur in 2a, where R2 is different. This interaction does not explain why no selectivity is observed in 2a.

Response: We thank the reviewer for the comment and agree that the schematic could be improved for clarity. We have revised the figure to primarily emphasize the synergistic interactions between the phenyl phosphate and the non-natural amino acid. This adjustment highlights how both electrostatic and π - π interactions contribute to the differences observed with **2b** and **3** in terms of coupling.

12) Figure 3, panel H: It is unclear why the authors define these aggregates of droplets/bubbles as multilayer. I suggest removing it because it is not apparent in the EM images.

Response: We have replaced the phrase multilayer bubbles with the word aggregates.

13) As mentioned, I found the last paragraph the most interesting of the manuscript, yet it is very superficial and not discussed. The covalent-self-sorting results are really clean and, if I understood well, suggest that **1e** does not "partition" in the aggregate with **1b**, so polymerization occurs in the solution. More explanation as to why this happens and how general it could be would be needed.

Response We appreciate the reviewer's recognition of the self-sorting phenomena and have further expanded on this in the revised manuscript. Our findings suggest that the self-sorting mechanism is closely linked to the aggregation behavior of the aminoacyl phosphate esters. Derivatives such as **1a**, **1d**, and **1g**, which do not aggregate, resulting in oligomerization occurring predominantly in the solution phase. In contrast, **1b** undergoes oligomerization in the aggregated phase, leading to a non-mixing environment and covalent self-sorting. Confocal microscopy further confirmed that **1b** formed spherical aggregates that transitioned into fibrillar structures over time, whereas **1g** remained in solution without forming aggregates. Fluorescence spectroscopy also supported these findings, as the emission spectra of mixed solutions containing **1b** and **1g** showed no change compared to **1b** alone, indicating that phase separation prevented interaction between the species. To further support this mechanism, we performed additional centrifugation experiments, which revealed the formation of two distinct phases, each selectively enriched in specific amino acid compositions. This observation reinforces the idea that phase behavior plays a critical role in covalent self-sorting.

This selective distribution between phases highlights the critical role of aggregation in promoting self-sorting and preventing the formation of mixed sequences. As suggested by the reviewer, the balance between hydrophobicity and hydrophilicity likely plays an important role in this selective phase behavior, which could be a general mechanism for systems where distinct side chain functionalities impact solubility and aggregation tendencies. The distribution of aminoacyl phosphate esters between the aggregated and solution phases is primarily governed by a combination of non-covalent interactions, including electrostatic forces, hydrophobic effects, and π - π stacking. Consequently, similar behaviors are observed in other systems featuring aliphatic side chains. We have expanded on these points in the revised manuscript.

We have updated the self-sorting section in the manuscript as follows: "Covalent self-sorting versus co-assembly: Building on our previous observations with N-terminus protected aminoacyl phosphate esters, we next explored peptide oligomerization using N-terminus free aminoacyl phosphate esters. We hypothesized that assembly of aminoacyl phosphate esters might allow for controlled oligomerization (Fig. 4A). Thus, we synthesized derivatives incorporating aromatic amino acid residues: FEP (**1a**), F(4-NH-Fmoc)EP (**1b**), F(4-guanidine)EP (**1e**), BPAEP (**1f**) and F(4-COOH)EP (**1g**). These activated amino acids were synthesized as ethyl esters. When **1b** and **1f** were mixed at equimolar concentrations (10 mM each) in 0.6 M borate buffer (pH 9.1), we observed the formation of mixed oligomeric species (Fig. 4B, Supplementary Figs. 115-117 and Supplementary Table 2). Microscopy studies revealed that both **1b** and **1f** formed spherical aggregates, similar to what was observed with previously studied protected aminoacyl phosphate esters like **2b** and **3** (Supplementary Figs. 118,119). Although no hydrogel formation was detected in the BPA-containing libraries, a weak hydrogel developed in the Fmoc-containing samples after oligomerization. Rheological studies further suggested that the incorporation of mixed sequences enhanced the mechanical properties of the resulting structures, pointing towards a co-assembly effect between oligomers (Supplementary Figs. 120, 121). The role of guanidine residues was also significant in these systems. When guanidine was present (**1e**), it promoted the formation of hetero-oligomers through electrostatic interactions with the phosphate esters, highlighting its importance in driving co-assembly and influencing the organization of mixed species (Supplementary Figs. 122-125 and Supplementary Table 3).

Next, we explored the behavior of libraries containing **1b** and **1g**. This mixture resulted in two distinct families of oligomers enriched in either Fmoc- or F(4-COOH)-containing species, indicating covalent self-sorting (Fig. 4C, Supplementary Figs. 126-128 and Supplementary Table 4). Centrifugation experiments revealed distinct phases, where the aggregated phase was predominantly enriched with **1b** oligomers, while **1g** oligomers remained in the solution (Supplementary Fig. 129). Confocal microscopy further confirmed that **1b** formed spherical aggregates that transitioned into fibrillar structures over time, whereas **1g** remained in solution without forming aggregates (Supplementary Fig. 130). Fluorescence spectroscopy also supported these findings, as the emission spectra of mixed solutions containing showed no change compared to **1b** alone, indicating that phase separation prevented interaction between the species (Supplementary Fig. 131). Self-sorting was also observed in libraries containing **1b** and **1a** (Supplementary Figs. 132-135 and Supplementary Table 5). To confirm the generality of this process, we extended the system to aliphatic aminoacyl phosphate esters, VEP (**1d**) (Supplementary Figs. 136-139 and Supplementary Table 6). Overall, these findings suggest that the assembly of aminoacyl phosphate esters creates a protective microenvironment that positions primary amines in close proximity to the acyl phosphate esters, facilitating selective coupling and homo-oligomerization. However, when additional non-covalent interactions, such as aromatic stacking (as seen with **1f**) or electrostatic forces (as observed with **1e**), are introduced, the aggregates engage in co-assembly, leading to the formation of hetero-oligomers. This balance between covalent self-sorting and co-assembly offers a versatile strategy for controlling peptide oligomerization, allowing specific amino acid residues to interact preferentially within different phases (aggregated vs. solution), ultimately driving the selective formation of oligomeric species with distinct chemical compositions.”

Next, we explored the behavior of libraries containing FmEP (**1b**) and F(4-COOH)EP (**1g**). This mixture resulted in two distinct families of oligomers enriched in either Fmoc- or F(4-COOH)-containing species, indicating covalent self-sorting (Fig. 4C, Supplementary Figs. 126-128 and Supplementary Table 4). Centrifugation experiments revealed distinct phases, where the aggregated phase was predominantly enriched with **1b** oligomers, while **1g** oligomers remained in the solution phase (Supplementary Fig. 129). Confocal microscopy further confirmed that **1b** formed spherical aggregates that transitioned into fibrillar structures over time, whereas **1g** remained in solution without forming aggregates (Supplementary Fig. 130). Fluorescence spectroscopy also supported these findings, as the emission spectra of mixed solutions containing **1b** and **1g** showed no change compared to **1b** alone, indicating that phase separation prevented interaction between the species (Supplementary Fig. 131). Self-sorting was also observed in libraries containing FmEP (**1b**) and FEP (**1a**) (Supplementary Figs. 132-135 and Supplementary Table 5). To confirm the generality of this process, we extended the system to aliphatic aminoacyl phosphate esters, VEP (**1d**) (Supplementary Figs. 136-139 and Supplementary Table 6). The results showed that the self-sorting mechanism can be applied across different aminoacyl phosphate esters, broadening the applicability of this approach. Overall, these findings suggest that the assembly of aminoacyl phosphate esters creates a protective microenvironment that positions primary amines in close proximity to the acyl phosphate esters, facilitating selective coupling and homo-oligomerization. However, when additional non-covalent interactions, such as aromatic stacking (as seen with BPAEP) or electrostatic forces (as observed with guanidine residues), are introduced, the aggregates engage in co-assembly, leading to the formation of hetero-oligomers. This balance between covalent self-sorting and co-assembly offers a versatile strategy for controlling peptide oligomerization, allowing specific amino acid residues to interact preferentially within different phases (aggregated vs. solution), ultimately driving the selective formation of oligomeric species with distinct chemical compositions.”

Fig. 4: Covalent self-sorting and co-assembly in spontaneous oligomerization reactions. A) Schematic representation of the oligomerization reaction between **1b** and other amino acyl phosphate esters (**1a**, **1d**, **1e**, **1f** and **1g**) resulting in covalent self-sorting or co-assembly. Formation of homo and hetero-oligomers between 10 mM **1b** with **B**) 10 mM **1f** and **C**) 10 mM **1g**. Libraries were prepared in 0.6 M borate buffer, pH 9.1. Peptide coupling was measured after 24 hours. Bar graphs represent the sum of all library components (grouped as homo or hetero-oligomers) formed in the reactions.

Additional figures in the Supplementary Information of the manuscript have been added.

Supplementary Figure 137: UPLC chromatograms of A) 10 mM **1b B), 10 mM **1d** and C) 10 mM **1b** with 10 mM **1d** (1:1), in 0.6 M borate buffer, pH 9.1.**

Supplementary Figure 130: Time-dependent confocal microscopy images of A) 10 mM **1b**. B) 10 mM **1g** in 0.6 M borate buffer, pH 9.1.

Supplementary Figure 131: Time-dependent Fluorescence emission spectra of A) 10 mM **1b** and B) Mixed library of 10 mM **1b** and 10 mM **1g** in 0.6 M borate buffer, pH 9.1.

Supplementary Figure 129: UPLC chromatograms of mixed library of 10 mM **1b** and 10 mM **1g** after centrifugation A) Oligomers of **1b** in aggregated phase B) Oligomers of **1g** in solution phase.

In summary, we have carefully considered all the points raised by the three reviewers and have made the necessary revisions to our manuscript accordingly. We greatly appreciate the valuable feedback provided by all the reviewers and we believe that the suggested changes have significantly enhanced the overall quality of our manuscript.

Response letter references

1. Dai K, et al. Spontaneous and Selective Peptide Elongation in Water Driven by Aminoacyl Phosphate Esters and Phase Changes. *J. Am. Chem. Soc.* **145**, 26086-26094 (2023).
2. Pol, M.D., Dai, K., Thomann, R., Moser, S., Roy, S.K. & Pappas, C.G. Guiding transient peptide assemblies with structural elements embedded in abiotic phosphate fuels. *Angew. Chem. Int. Ed.* 2024, e202404360 (2024).
3. Tamura, K. & Schimmel, P. Peptide synthesis with a template-like RNA guide and aminoacyl phosphate adaptors. *Proc. Natl. Acad. Sci. U.S.A.* **100**, 8666–8669 (2003).
4. Carpenter, F. H. The Free Energy Change in Hydrolytic Reactions: The Non-ionized Compound Convention. *J. Am. Chem. Soc.* **82**, 1111–1122 (1960).
5. Thornton K, et al. Mechanistic insights into phosphatase triggered self-assembly including enhancement of biocatalytic conversion rate. *Soft Matter*, **9**, 9430-9439 (2013).
6. Lander A.J., et al. Roles of inter- and intramolecular tryptophan interactions in membrane-active proteins revealed by racemic protein crystallography. *Commun. Chem.* **6**, 154 (2023).